# Framework for the Simulation of Sensor Networks Aimed at Evaluating In Situ Calibration Algorithms

**DOI:** 10.3390/s20164577

**Published:** 2020-08-14

**Authors:** Florentin Delaine, Bérengère Lebental, Hervé Rivano

**Affiliations:** 1Efficacity, F-77420 Champs-sur-Marne, France; berengere.lebental@univ-eiffel.fr; 2COSYS-LISIS, Univ Gustave Eiffel, IFSTTAR, F-77454 Marne-la-Vallée, France; 3LPICM, CNRS, Ecole Polytechnique, Institut Polytechnique de Paris, F-91128 Palaiseau, France; 4Université de Lyon, INSA Lyon, Inria, CITI, F-69621 Villeurbanne, France

**Keywords:** sensor network, calibration, simulation, evaluation

## Abstract

The drastically increasing availability of low-cost sensors for environmental monitoring has fostered a large interest in the literature. One particular challenge for such devices is the fast degradation over time of the quality of their data. Therefore, the instruments require frequent calibrations. Traditionally, this operation is carried out on each sensor in dedicated laboratories. This is not economically sustainable for dense networks of low-cost sensors. An alternative that has been investigated is in situ calibration: exploiting the properties of the sensor network, the instruments are calibrated while staying in the field and preferably without any physical intervention. The literature indicates there is wide variety of in situ calibration strategies depending on the type of sensor network deployed. However, there is a lack for a systematic benchmark of calibration algorithms. In this paper, we propose the first framework for the simulation of sensor networks enabling a systematic comparison of in situ calibration strategies with reproducibility, and scalability. We showcase it on a primary test case applied to several calibration strategies for blind and static sensor networks. The performances of calibration are shown to be tightly related to the deployment of the network itself, the parameters of the algorithm and the metrics used to evaluate the results. We study the impact of the main modelling choices and adjustments of parameters in our framework and highlight their influence on the results of the calibration algorithms. We also show how our framework can be used as a tool for the design of a network of low-cost sensors.

## 1. Introduction

In recent years, the emergence of low-cost environmental sensors has raised a strong interest, such as those for air pollution monitoring in cities [1,2]. They enable the deployment of large sensor networks at reasonable cost to provide information at high spatial resolution [3].

However, low-cost monitoring technologies usually suffer from various problems that still need to be tackled. One of them concerns the degradation of quality of the measured values over time [3,4]. Studies are frequently evaluating the performances of such devices from this point of view [5,6,7]. More precisely, low-cost sensors are subject to drift. This can be mitigated with periodic calibration [8]. Traditionally, calibration of a sensor is carried out in a controlled facility (e.g., in a laboratory), where it is exposed to standard values or co-located to a reference instrument. This operation, even when conducted in field [9], is often not affordable in the context of dense low-cost sensor networks. To mitigate this cost, various in situ calibration strategies have been developed [10,11,12]. With them, calibration is carried out while leaving the sensors deployed in the field, preferably without any physical intervention.

Usually, the performances of calibration strategies are evaluated on case studies where a reference dataset is compared to faulty measurements corrected after the calibration. The case studies are either based on experiments with real data from actual sensors under laboratory or field conditions [13], or on computer simulated values of the phenomena and of the sensors [14]. Unfortunately, performing inter-comparison between these strategies is challenging. Each one is evaluated on its own case studies [12] and there is no commonly used test scenario in the literature beyond some toy examples. Besides, the settings of each case study can be complex and difficult to replicate. In short, the literature is missing a systematic benchmark for calibration algorithms [10,11].

This paper addresses this question. Our contribution consists of a generic framework to design case studies that can be used for quantitative comparisons of in situ calibration strategies. Its objective is to get a better understanding of the factors driving the performances of calibration strategies and the quality of sensor networks. For reproducibility and scaling issues, it is based on numerical simulations of the environmental phenomena and of the sensors, their metrological properties as well as the faults introduced in the measurements. It is henceforth not a new calibration methodology, but the first methodology to carry out a systematic inter-comparison of the performances of existing methodologies.

We apply this framework on seven calibration algorithms. In a first step, we consider an elementary case study on a blind static network that is sufficient to get first insights by analysing the performances with commonly used metrics at different scales. Then, we evaluate the impact of the realism of the model of pollution emission and dispersion, the density of the deployment of the sensors, and the model of error of the sensors. Our investigations provide the following engineering insights.

There is actually no method that is universally outperforming the other. The best calibration strategy to apply depends on the deployment of the sensors and the goal of the measurement.Even on the same scenario, two strategies can outperform each other depending on which metric of performance is considered and how it is averaged.Increasing the density of the deployment of sensors does not always lead to better performances of the calibration strategy.In some case, the quality of data can be degraded by a calibration strategy.

This validate the relevance of such a framework in the design phase of a practical network of low-cost environmental sensors. In addition, we provide the codes and data we used in this work as Appendix A.

Roadmap. In Section 2, we review the challenges of the comparison of in situ calibration strategies. The framework we propose is presented in detail in Section 3. An elementary case study is carried out in Section 4, showing how to put the methodology in practice. In Section 5, we investigate in more details how the choice of metrics and their representation can change the perception of the previous results. We also propose an error model-based approach for performance evaluation. Section 6 tackles how changes in the different parts of the framework can influence the results of a case study. A discussion on the use of the framework based on the conclusions made in the previous sections is carried out in Section 7, followed by a general conclusion.

## 2. Related Work

In publications introducing new methods for in situ calibration, case studies are systematically conducted to demonstrate how to put the algorithm in practice and its efficiency. This step is either performed through simulations, laboratory experiments or field experiments.

The principle consists in comparing the measured values, both before and after calibration, to the values that should have been measured by the instruments if they were ideal. The latter values are obtained with simulations or with instruments of a higher quality, co-located to instruments to calibrate.

Different approaches by simulation have been proposed to define the values of the measurand. It can be based on 2D Gaussian fields [15] or ARMA processes [16] for instance. Disturbances like noise or drifts are added to the reference values to generate the measured values. However, the models are more or less realistic, questioning the relevance of the results obtained, with respects to experiments.

Concerning experimental approaches, there are lots of datasets produced and used in various studies (for instance the Intel Lab Data [17], a deployment at James Reserve used in Reference [18], the OpenSense Zurich Dataset [19]). They may be representative of particular types of deployments, for instance indoor (Intel Lab Data) or outdoor (James Reserve, OpenSense) measurements, with static and dense sensor networks (Intel Lab Data, James Reserve) or mobile sensor networks (OpenSense). Moreover, whereas repeating a simulation with different network settings (number of instruments, types of instruments, positions...) but identical phenomenon is feasible, it is almost impossible with experiments except under laboratory conditions. Although there are existing facilities that could allow it [2], they are not adapted for the study of sensor network over large areas. A solution could be the deployment of as many instruments as necessary to produce the desired configurations, but this would increase the cost of the experiment in an unrealistic manner. Moreover, this would have to be reproduced in multiple environments to study their influence on the performances of in situ calibration strategies.

More generally, whether we focus on simulation-based or experimental approaches, there is no single case study that is reused a significant number of times in multiple publications and by different groups of authors [12]. For simulation-based ones, they are not always easily reproducible, and the choice of parameters is often left undiscussed. Regarding experiments, datasets are often shared with the community, though they sometimes become unavailable over time or require post-processing that is left to the user. It does not facilitate the comparison of existing works. Although, such an effort of comparison is conducted in Reference [20] for instance. The performances after calibration of measuring systems using sensors from the same manufacturer are compared based on the results published by different groups of researchers. The drawback is that results were not obtained on the same case study. In this way, the performances are not really comparable.

Also, in the perspective of comparative studies, few codes are open-sourced. Therefore, algorithms have to be fully reimplemented in most cases to achieve comparison [21] and due to the efforts it requires, it is carried out in comparative studies only for strategies with features similar to the one they propose–for example the use of machine learning techniques [22,23,24] or the use of Kalman filter [21].

Overall, while a new strategy can be over-performing previous ones on a given case study, one rarely knows whether the new strategy is better in general or only for the specific case, regardless of whether it is based on simulation or on experiment. Thus, there is a strong need for systematic tools and protocols [10,25] enabling to compare the performances of in situ calibration methodologies across studies.

## 3. Description of Our Framework

A generic framework to design case studies is proposed in this section for quantitative comparisons of in situ calibration strategies. It aims at yielding a better understanding of the factors driving the performances of calibration strategies, and at providing a protocol as rigorous as possible to conduct such studies. The different steps are described in a generic way to ensure their applicability for most cases.

In an effort of having the most consistency with the vocabulary of the International Vocabulary of Metrology (VIM) [8] applied to laboratory or field experiments, several terms that we use afterwards are defined below. The formal definitions of terms in bold are available in [8].

In our framework, we adopt a data-centric point of view. We do not target the modelling of the entire measuring chain. We assume that instruments are grey boxes which provide **measured values** based on **true values** of a **measurand** and possibly also on true values of **influence quantities**. Therefore, the notion of **indication** is hidden. The grey boxes representing instruments consist into algorithms mimicking the features of real instruments. This is illustrated in Figure 1.

The formal definition of **calibration** is the matching of indications with standard values–and their associated uncertainties–and the derivation of a relationship to infer measured values from indications. In situ calibration strategies aim at providing for an instrument a relationship to obtain corrected measured values from measured values, with the help of the measured values of other instruments.

In this work, we focus on the metrological performances of in situ calibration strategies. Other subjects such as the communication costs, the energy consumption or the computational efforts for instance are not in the frame of this study.

### 3.1. Simulation-Based Strategy

Our methodology is based on simulation because it enables the following properties that are difficult to get with field or lab experiments:Ability to perform a study with different operating conditions for a same sensor network.Reproducibility: same operating conditions on different sensor networks.Knowledge of true values: in experimental operating conditions, true values are not known perfectly–there is always an uncertainty– and having very accurate values requires high quality instruments which are usually expensive.Scalability: the only limit to the density, number and diversity of sensors is the computing time.

To conduct the evaluation of in situ calibration strategies, the framework combines the simulations of the environmental phenomenon to produce the true values of the measurand, and of the eventual influence quantities, at any position and time, the simulations of the mobility –if any– of the nodes of the network to know where and when measurements are performed, and the simulations of the measuring chain of the instruments to produce the measured values. The realism of the results will depend on the complexity and accuracy of each simulation model. On the other hand, being able to analyse results on a simplistic model can also help to highlight fundamental properties before confronting them to more complex situations. In particular, even if we consider networks of sensors, we neglect to simulate the system and networking aspects. We consider they have a lesser influence on the metrological performances of in situ calibration strategies. One could however argue that system and network issues could challenge the robustness of a given implementation of a calibration protocol because of packet loss. It could be also interesting to evaluate the energetic cost or time to converge of such implementation. It is still possible to extend the framework to these issues that could be covered by tools such as WSNet (http://wsnet.gforge.inria.fr/), NS-3 (https://www.nsnam.org/) or other (OpNet, LoRaSim….).

The objective of this framework is to help end-users in the design of their sensor networks by:showing which strategies are applicable to a given use case. Currently only a few strategies are supplemented with strong formal criteria enabling to determine whether they can be used for a given network and in a given situation, for instance in Reference [14] or Reference [18]. Such criterion may however be not easy to define and therefore simulation is an interesting solution.showing which strategies should give the best performances regarding the assumptions that can be made in practice.allowing the optimization of the settings of the calibration strategies –as in Reference [13] where the time period of calibration is studied. As there are rarely formal applicability criteria, there is also rarely protocols defined for the adjustment of the parameters.

### 3.2. Functional Decomposition

Our methodology, represented in Figure 2, can be described schematically as follows:Build a dataset of ideal measured values.(a)Simulate the quantities involved with a sufficiently high resolution (spatially and temporally).(b)Simulate the positions of the sensors.(c)Combine the two simulations.Build measured values from the ideal measured values for example, add defects to these values.Perform calibration for example, determine the correction to apply to the measured values.Derive corrected measured values.Compare measured and corrected measured values with ideal measured values. If the algorithm performs well, the corrected values should be closer to the ideal values than the measured ones.

To apply this methodology, input data are the number of instruments involved, the quantities measured, the in situ calibration strategies to apply and the metrics considered for the evaluation of the results but also:Information to simulate the quantities involved. Depending on the model, it can be as simple as some parameters of an equation (e.g., a Gaussian model) or a complex description of an urban scene (e.g., http://air.ec-lyon.fr/SIRANE/)Specifications on the deployment and possible mobility of the instruments. Since the positions of the instruments is critical for the performances of the sensor network, the number of instruments deployed, and their positions may be optimized [26].Descriptions of the measuring chain of the instruments involved, in terms of drift and of their eventual influence quantities.Configuration of the considered in situ calibration strategies.

As outputs, the framework provides time series of the true values, of the measured values and of the corrected values for each instrument. Values of metrics computed for the comparison of these times series may also be given.

The following subsections give details on each step.

#### 3.2.1. Build Ideal Measured Values

Ideal measured values are measured values that should be measured by instruments if they were perfect. It means that for an instrument $s_{i}$, its measured values $v(s_{i},t)$ are equal to the true value of the measurand, noted $v_{true}\left(s_{i},t\right)$, for all *t*.

Generating these ideal measured values for all the instruments of a sensor network can be carried out as follows.

To build ideal measured values, true values of the measurand must be computed. True values of other quantities may also be derived for the simulation such as influence quantities if the instruments to model undergo such effects.

To be able to repeat the study with different sensor networks, the quantities can be simulated independently of the positions of the instruments so that one may build different sets of ideal measured values from a same simulation of the quantities depending on the sensor network.

Therefore, the simulation is performed on the targeted domain of study for example, a specific geometry, at a sufficient spatial and temporal resolution, regarding the potential positions of the instruments and their frequency of measurement. The model used for the simulation depends on the application and on the targeted accuracy. Ideally, the simulation should perfectly represent the studied environmental phenomenon and the quantities affecting instruments operating under real conditions. This is not always possible and thus the models used may not reflect the real temporal and spatial variability of environmental phenomenon. This is particularly true for air pollution monitoring. While high model accuracy can be achieved with advanced computational fluid dynamics models [27], such models require time and resources that are not always acceptable or available for users. In multiple works, case studies were conducted with abstract models [18], enabling to demonstrate that the proposed calibration strategy is functional. However, such models do not enable determining if an algorithm is applicable on real cases.

In addition to the true values of the considered quantities, the sets of the positions where measurements are performed must be built with respect of the position and possible mobility of each instrument.

Based on the positions of the instruments over time, true values are extracted and stored from the simulation of the studied quantities at the locations of each instrument at each time step according to their sampling periods (In this section, we considered that the simulation of the quantities and the simulation of the positions of the instruments are carried out separately and then combined. However, it might possible to determine first the positions of the instruments and then to derive the ideal measured values only at the positions for instance).

Afterwards, a time series of ideal measured values is available for each sensor. After this step, these time series must be validated against the hypotheses of the targeted calibration strategies.

#### 3.2.2. Build Measured Values from the Ideal Measured Values

Until this step, sensors are assumed ideal. Thus, to obtain realistic measured values e.g., values that include the real behaviour of measuring instruments, faults must be introduced. The term “fault” is used according to its definition in the field of dependability [28]. It is the cause of an error generating the failure of the system for example, its deviation from its correct service. The correct service here is the delivery of measurement results satisfying the metrological specifications of the instrument. The fault addressed by calibration strategies is the instrumental drift. It is defined according to the VIM as a “continuous change over time in [measured values] due to changes in metrological properties of a measuring instrument.” This is one of the most complex faults to detect. The influence on the performances of in situ calibration strategies of other faults (e.g., noise, spikes, missing data...) can also be studied by introducing them in the measured values (see Section 6.3.2).

#### 3.2.3. Perform Calibration and Build Corrected Values

Once measured values are obtained, calibration may then be performed low-cost to correct them. It must be based on algorithms applicable to the specific case study addressed.

Calibration and correction are separated in the methodology for the sake of genericity. Some calibration strategies directly determine the correction to apply to each value [29], while others provide parameters of a mathematical relationship to apply to each measured value [18].

#### 3.2.4. Evaluate

Finally, corrected values, measured values and true values can be compared to assess the performances of the algorithms. Most of the time, usual metrics are employed such as root-mean-square error, mean absolute or Pearson’s correlation coefficient for each sensor in the network. Results of case studies may also be presented as the mean of these metrics over the whole network [30]. We discuss the most suitable metrics to use in Section 5.

In the following sections, the framework is applied to the comparison of several strategies for blind static networks on a case study that is simple enough to focus on fundamental issues.

## 4. Comparison of In Situ Calibration Strategies for Blind Static Sensor Networks

### 4.1. Frame of the Study

In this section, we address blind static sensor networks. All the instruments remain at the same position and no reference instrument is present. This type of sensor networks offers a particular challenge in terms of calibration. By contrast, when reference instruments are in the network, trustworthy values are available. Secondly, when instruments are mobile, it is possible that two or more instruments are measuring the same true quantity value when they are in a spatio-temporal vicinity. For blind static sensor networks, the availability of standard values may not be assumed, and only the instruments that are deployed close enough can compare their measurements.

We consider seven calibration strategies, from four different groups of authors, that were identified in a previous work [12]. These strategies have never been compared all together on a common case study.

Balzano et al. [18]: To apply this algorithm, the sensor network must be dense enough to oversample the signal of interest. We refer to this assumption by the term “oversampling” afterwards. With such a sensor network, the true values lie in a subspace of the space formed by the measured values. To calibrate the instruments, this subspace must be known. The gains and offsets for all nodes are then estimated with the chosen subspace and the measured values using singular value decomposition (SVD) or by solving a system of equations using a least square (LS) estimator. These strategies are respectively called **SM-SVD** and **SM-LS**, SM standing for subspace matching. Note that gains are estimated up to a multiplicative coefficient. Consequently, the gain of one instrument should be known to complete the calibration. Likewise, the offset computation requires either to know some of the offsets, or that an additional hypothesis on the signal is valid. (A usual assumption is that “the signal has a null (or known) average”.) As such, the network is not perfectly blind in truth.Lipor et al. [31]: This work is an extension of Reference [18]. It shows first that the solution obtained with SM-SVD is equivalent to the one obtained by solving the system of equations expressed as a total least squares problem (TLS). The latter strategy is called **SM-TLS**. (Note that additional formulations given in the publication, in the case where some gains are known for a subset of instruments, are not considered here as it would be equivalent to have a partially blind sensor network.)Takruri et al. [29,30,32]: Three contributions from these authors are considered. The first one is based on the idea that the average of the values of neighbouring sensors gives a good estimation of the correct value of a sensor to calibrate. If the absolute difference between the sensor value and the average is greater than a given threshold, this difference is added to the sensor’s value. We call this strategy **AB-DT** (average-based estimation for difference-based threshold). Two variations are also proposed, a first one in which the difference-based threshold mechanism is replaced by a Kalman filter that estimate the error affecting the instruments (**AB-KF**), and a second one where the average-based part is replaced by support vector regression (**SVR-KF**).Kumar et al. [33,34]: This work, inspired from the previous one, also propose the use of a Kalman filter to estimate the error of each instrument but instead of performing an average over neighbouring nodes or using SVR for true value estimation, kriging is used (**K-KF**).

Note that not every strategy that could have been applied to our case study is considered here. For instance, the work of Dorffer et al. [35] extends Reference [18] and Reference [31] for sparse signals, but this is not in the scope of the study.

### 4.2. Application of the Framework

#### 4.2.1. Simulation of the Ideal Measured Values

A space of 1000×1000m is considered. It is discretized with a step of 10 m. At the center of this area, a $NO_{2}$ source is considered. The concentration of $NO_{2}$, *C*, at the instant *t* and position (x,y), is modelled as:$$C\left(x,y,t\right)=A\left(t\right)\exp\left(-\frac{x^{2}+y^{2}}{\sigma {\left(t\right)}^{2}}\right)$$

It is a 2D Gaussian function with an equal spread σ for *x* and *y*. This model is not very realistic but has been used in other papers [14] for its simplicity of implementation. Examples of pollution maps are represented in Figure 3.

To facilitate the interpretation of σ, it is expressed as a function of the full width at half maximum (FWHM) of the Gaussian curve:$$\sigma =\frac{FWHM}{2\ln(2)}$$

*A* and FWHM are functions representing respectively the temporal evolution of the amplitude of the 2D Gaussian function and of its FWHM.

To be representative of actually measured concentrations, the function *A* is based on values measured by a $NO_{2}$ monitoring station in Paris between 2017/01/01 00:00 and 2017/12/31 23:00, with an hourly time step (Paris centre station [36]). (There were missing entries in the time series. Values were first interpolated with a limit of 3 consecutive values to fill, for example, 3 h. Then, in the case where more than 3 consecutive values are missing, the interpolation is made based on the values at the same hour of the previous and next day. The values are interpolated with a linear function for both cases.) We consider it represents our pollution source at the coordinates (0,0). Even if the values of the station are not the ones of a real source, we assume it gives a reasonable order of magnitude of variation over time.

The FWHM represents the spread of the pollutant around the source. Two linear piecewise functions are defined to represent the daily evolution of the FWHM: one for weekdays and one for weekends. They are represented in Figure 4b. Their shape is inspired from the dataset used for the amplitude *A*. This is not fully realistic, but it provides an order of magnitude of spatial and temporal variations occurring in urban context.

The concentration of pollutant *C* is simulated for a year at an hourly time-step over the area of study. To avoid the computation of *C*, at each time step, for the current value of *A* and FWHM, a catalogue of maps is generated from sets of possible values for the amplitude and the FWHM: respectively from 0 to 150 μg · m^−3^ with a step of 5 μg · m^−3^ and from 0 to 4000 m with a step of 500 m. In our case, it required the simulation of 157 maps. To derive the time series, the map generated with the closest allowed values of *A* and FWHM is picked from the catalogue at each time step. Finally, a time series of 8760 maps is obtained.

A sensor network *S* of |S|=16 static measuring instruments is considered. It is uniformly deployed spatially. The positions are represented in Figure 5. Then, the time series of concentration maps and the positions of the instruments are combined to obtain the time series of ideal measured values for all the instruments, noted $v_{true}$. In this case, we consider that the instruments measure one value per hour.

The validity of this set of ideal measured values must be verified for SM-(SVD,LS,TLS) strategies. The network should satisfy the hypothesis of “oversampling”. To verify it, a principal component analysis (PCA) of the matrix formed by the concatenation of the ideal measured values time series of each node is carried out. We assume that the sensor network is satisfying the oversampling hypothesis if the explained variance ratio of the ν first components is greater than 0.999 and ν<|S|. In our case, the condition is met for ν=2<|S|. Therefore, the oversampling hypothesis is satisfied.

#### 4.2.2. Building of the Measured Values

The network is defined as “blind”, which can also be expressed by “all the instruments have the same sensing reliability”. In a first step, instruments are only drifting. Drift of each instrument is only function of time. We assume all the instruments undergo a linear drift increasing up to 5% of the gain every 7 days. The actual drift is randomly drawn following a uniform law. The value of 5% is set according to what is reported in typical sensor datasheets [37].

The sensors are assumed to be initially calibrated and to remain faultless for eight weeks, which gives a training dataset for the strategies needing it, notably to learn the subspace of the signal with PCA for SM-(SVD,LS,TLS). For four additional weeks, the sensors are kept without drift to study how calibrations strategies behave if instruments are not drifting. Finally, sensors start drifting after twelve weeks. This instant is noted $t_{\mathrm{start}\;\mathrm{drift}}$.

Therefore, the gain $G(s_{i},t)$ of the instrument $s_{i}$ at *t* is:$$G\left(s_{i},t\right)=\left\{\begin{array}{cc} 1 & \mathrm{if}\;t<t_{\mathrm{start}\;\mathrm{drift}}\\ G(s_{i},t-1) & \mathrm{if}\;t\geq t_{\mathrm{start}\;\mathrm{drift}}\;\mathrm{and}\;\left(t-t_{\mathrm{start}\;\mathrm{drift}}\right)\;\mathrm{mod}\;7\;\mathrm{days}\;\neq 0\\ G\left(s_{i},t-1\right)+\delta G\left(s_{i},t\right) & \mathrm{if}\;t\geq t_{\mathrm{start}\;\mathrm{drift}}\;\mathrm{and}\;\left(t-t_{\mathrm{start}\;\mathrm{drift}}\right)\;\mathrm{mod}\;7\;\mathrm{days}\;=0 \end{array}\right.$$
$$\;\mathrm{with}\;\forall t,\delta G\left(s_{i},t\right)∼U\left(0,0.05\right)$$

This drift model is called “Weekly Gain Linear Increase” (WGLI).

The measured value $v(s_{i},t)$ of the instrument $s_{i}$ at *t* is expressed from the true value $v_{true}\left(s_{i},t\right)$ using the relationship:$$v\left(s_{i},t\right)=G\left(s_{i},t\right)\;\cdot \;v_{true}\left(s_{i},t\right)$$

#### 4.2.3. Configuration of the Strategies

Each strategy has parameters to set up. They are defined as follows:SM-(SVD,LS,TLS): All these strategies share the same parameters, namely the periodicity of calibration and the time interval on which the linear system is solved. Note that the tuning of the parameters was not discussed by the original authors. We chose to apply the strategy each week and to solve the linear system over the past w=7 days.Kalman filter of (AB, SVR, K)-KF: Variables *R* and *Q* are set to 0.0001 and 1 respectively.Definition of neighbours for AB-(DT,KF): Sensors are defined as neighbours if they are distant of less than 250 m.Parameters of the SVR for SVR-KF: Kernel used is ’rbf’, the kernel coefficient gamma is set to 10^−8^ and the penalty parameter *C* is set to 10^3^.Parameters for the weighting for K-KF: *a*, $c_{0}$ and $c_{1}$ were defined as in Reference [34], therefore a=12, $c_{0}=0.25$, $c_{1}=0.85$.

We invite the reader to refer to the original publications for a more in depth understanding of the parameters of these calibration strategies.

#### 4.2.4. Evaluation

To compare the corrected values of an instrument, obtained with each strategy, and its measured values against its ideal measured values, metrics are needed. We consider the most commonly used metrics in the publications reported in Reference [12] and described as follows. They can be computed for each instrument or averaged over the network. We discuss the relevance of these statistics in Section 5. In the following, the set *x* stands for the *k* measured values of an instrument $s_{i}$ over [t−Δt;t], $V(s_{i},\left(t,\Delta t\right))$ or the corresponding set of corrected values $V_{corr}\left(s_{i},\left(t,\Delta t\right)\right)$ and the set *y* is the associated *k* true values $V_{true}\left(s_{i},\left(t,\Delta t\right)\right)$.
Mean absolute error (MAE):
$$MAE\left(x,y\right)=\frac{1}{k}\sum_{i=0}^{k}\left|x_{i}-y_{i}\right|$$Root-mean-square error (RMSE):
$$RMSE\left(x,y\right)=\sqrt{\frac{1}{k}\sum_{i=0}^{k}{\left(x_{i}-y_{i}\right)}^{2}}$$Mean absolute percentage error (MAPE):
$$MAPE\left(x,y\right)=\frac{100}{k}\sum_{i=0}^{k}\frac{|x_{i}-y_{i}|}{|y_{i}|}$$Remark: the values of *y* must all be different from zero.Pearson correlation coefficient (ρ):
$$\rho \left(x,y\right)=\frac{E[(x-E[x])(y-E[y])]}{\sqrt{E\left[{\left(x-E\left[x\right]\right)}^{2}\right]E\left[{\left(y-E\left[y\right]\right)}^{2}\right]}}$$
with E[.] being the expectation.

For perfect instruments before correction (or after with $v_{corr}$ instead of *v*):
$MAE(V\left(s_{i},\left(t,\Delta t\right)\right),V_{true}\left(s_{i},\left(t,\Delta t\right)\right))=0$.$RMSE(V\left(s_{i},\left(t,\Delta t\right)\right),V_{true}\left(s_{i},\left(t,\Delta t\right)\right))=0$.$MAPE(V\left(s_{i},\left(t,\Delta t\right)\right),V_{true}\left(s_{i},\left(t,\Delta t\right)\right))=0\%$.$\rho (V\left(s_{i},\left(t,\Delta t\right)\right),V_{true}\left(s_{i},\left(t,\Delta t\right)\right))=1$.

### 4.3. Results

In Figure 6, a partial plot of true values, measured values and corrected values obtained with each strategy is displayed for a particular sensor. For the considered sensor, a visual observation indicates that strategies SM-(SVD, LS, TLS) and SVR-KF provide better results than AB-DT, AB-KF and K-KF.

Despite its advantage to easily visualize the results, this representation is not representative of all the instruments of the network.

The computation of the mean and standard deviation of each metric over the whole sensor network, on the entire time interval of study and with the results of each strategy, plus without calibration, is given in Table 1. With it, we observe that:SM-(SVD, LS, TLS) strategies have the best results overall with a small mean error and standard deviation whatever the metric consideredSVR-KF which seemed to give interesting results in Figure 6 provides corrected values only slightly better than before calibration according to MAE and RMSE but not according to MAPE and Pearson correlation coefficient. This could be explained by two reasons. The strategy may correct well for high measured values but correct poorly for low ones. It could also be due to errors in the corrected values of particular sensors introduced by the calibration algorithm. The aggregation by averaging makes it impossible to discriminate between these two possible explanations.AB-(DT, KF) and K-KF do not improve the measured values

These two last observations invited us to look at results for particular instruments. Table 2 gathers the results for two sensors selected randomly. We observe that strategies AB-(DT, KF) and (SVR, K)-KF are quite equivalent for these two instruments. The improvements are rather small for both sensors. For SM-(SVD, LS, TLS), results are consistent with our previous observation based on the average and the standard deviation in Table 1. The standard deviation does give the information of the existence but not the identity of the sensors that are degraded or improved by calibration, which is needed for practical deployments to drive the maintenance operations.

Furthermore, note that the results of Table 1 were computed over the 12 weeks of drift. The results may be different if computed over a different time range, for instance over each week. Figure 7 shows the evolution over time of the MAE computed each week for a particular instrument and each strategy. Table 3 provides statistics on the MAE computed each week but also for the other metrics that were computed in the same way. From the standard deviations of MAE, MAPE and RMSE, it shows that the observations made previously could be locally false, e.g., a strategy is better than others considering a computation of the metrics over the 12 weeks of drift but not always considering a computation of the metrics over each week. This is shown in Figure 7 where results with MAE for SVR-KF are nearly always worse than those for AB-DT, AB-KF and K-KF until week 24 but are better afterwards. This figure also shows that the performances of SVR-KF could be explained by the presence of a bias, at least according to MAE because it is quite constant in Figure 7.

### 4.4. Conclusions

Through this evaluation, we have shown that existing in situ calibration strategies could improve the measured values of a blind static sensor network in the considered case study. Overall, the strategies SM-(SVD, LS, TLS) have the best performances. This can be explained by the fact that the gain of at least one of the instruments in the network has to be known for these methods. In a way, the sensor network is only partially blind, but acquiring this information may not be a major issue in practice for large sensor networks. The other strategies appear to be able to mitigate the drift by few per cents at most only in the best cases, depending on the metric considered. The comparison between metrics enables understanding better how they operate. For all algorithms, studying the average values of the metrics over all the network hides strong disparities from sensor to sensor.

However, these results could be challenged based on the following considerations:The results may be different with another network (different positions and increased number of instruments, notably for AB-DT, AB-KF and K-KF strategies that use values of neighbour sensors for the prediction of correct values). However, the network is often defined by the use case. Thus, it is not always a criterion that can be optimized.The model of the considered phenomenon is not realistic and therefore results may not be practical. It is however sufficient to highlight issues of the algorithms under test. A more realistic model may not yield better result but would certainly make the analysis more complex.Only a drift of the gain was added which does not fully relate the behaviour of a real instrument. Adopting a finer-grained model of the behaviour also increases the complexity of the case study.Other parameters of the calibration strategies may change the results. We empirically explored the parameters of the algorithms, keeping in this study the values that give the best results for each strategy. There is however no claim for optimality here, but the framework can be used to optimize the parameters of a given strategy.

In Section 6, we test the impact of these four separate aspects on the algorithms. Before moving on to this, we discuss in the next section the metrics used and how additional ones, or other ways to visualize results, can give an enhanced comprehension of the behaviour of the calibration algorithms on a sensor network.

## 5. Evaluation of Measurements after Correction

### 5.1. Problem Statement

To evaluate the performances of in situ calibration strategies, various metrics are commonly used like MAE, MAPE, RMSE and Pearson’s correlation coefficient notably. We presented the results as statistics of these metrics over the entire network or values for particular instruments. The metrics were computed on the entire time interval of study or on sub-intervals. Results were displayed as tables and plots. These choices enabled carrying out an analysis at multiple scales (entire network versus individual sensor). This approach underlines disparities of efficiency of the algorithms on particular sensors for instance.

Multiple surveys exist on metrics [38]. They aim at providing a clear reading on the differences and similarities among the wide spectra of metrics used. While metrics are numerous, groups of them are based on the same idea (difference, absolute difference, square difference...). In some works, such as in Reference [39], an integrated performance index was developed to combine the values of several metrics into a single one. However, some metrics are interdependent [40], which can exaggerate or hide information when they are combined. In the literature regarding in situ calibration strategies, the choice of metrics is not always clearly justified, nor is the choice of the values presented and how they are computed (values for particular instruments, statistics of the metric over the network and so on). Moreover, Tian et al. [40] showed it is possible to have equal values of multiple metrics for two sets of values very different from each other. They justified it by the interdependence, underdetermination and incompleteness of usual metrics, in addition to the fact that they assume the error to be linear. This issue is major as metrics are used in official recommendations such as the data quality objectives (DQO) in the context of air quality measuring instruments [7,10,41]. In this way, we suggest in the next section to use an error model as introduced by Tian et al. [40]. It is considered as a satisfying mean to deal with these issues.

Another question lies in the way a metric is computed or in how statistics are derived from it. As shown in the previous section, looking at values of metrics computed over the full-time interval of drift or computed over each week may yield different interpretations. The choice of the time interval on which it is derived can be guided by requirements on the minimal duration for which the mean error of an instrument must be under a given threshold for instance.

Regarding the use of statistics, it concerns particularly networks with hundreds of nodes, for which interpreting the values of metrics for each node may not be easily feasible. However, statistics derived over the entire network do not enable discriminating easily between sensors differently affected by an algorithm. The use of statistics is perfectly appropriate if the quality of the calibration of the instruments belonging to a sensor network should be evaluated at the level of the network, for example, when it is the collective performance that matter. Otherwise, if each instrument is expected to be perfectly calibrated after an in situ calibration, statistics may only give a first idea of the performances of the algorithm. At the same time, evaluating the performances of an algorithm on a very large sensor network by looking at each sensor individually may not be feasible. Thus, the scale at which the performances of the instruments are evaluated must be specified in the description of a study, to make a proper use of statistics and to determine if the analysis is manageable in practice.

In addition, the way the results are displayed is important. We discuss this topic in Section 5.3 and propose the use of an easily readable representation, compatible with the error model we use hereafter.

### 5.2. Evaluation with an Error Model

We investigate in this section an approach which captures better individual sensor behaviours. It applied to the previous case study for the sake of comparison. We follow the error model approach proposed by Tian et al. [40]. It consists in finding a function F associating each measured value of an instrument $s_{i}$, noted $v(s_{i},t)$, to the true values of the measurand $v_{true}\left(s_{i},t\right)$ so that $v\left(s_{i},t\right)=F\left(v_{true}\left(s_{i},t\right)\right)$. (This idea of error model is close to the concept of a calibration relationship. The inverse function of the error model is the relationship to apply to an instrument so that calibration is perfect.) They illustrated this with a linear additive error model expressed as:$$v\left(s_{i},t\right)=a\;\cdot \;v_{true}\left(s_{i},t\right)+b+\varepsilon \left(t\right)$$
with *a* and *b* being respectively a constant gain and a constant offset, and ε a random error, following a given probability law depending on the case. If the actual distribution of ε is not determined, it is equivalent to the fitting a linear trend model between *v* and $v_{true}$. A linear regression can be used to determine the slope *a* and the intercept *b* in this case. a=1 and b=0 are the ideal results, indicating a perfect calibration. The linear regression between *v* and $v_{true}$ is appropriate to describe the performances before calibration. Comparing the slope and intercept of this regression to the ones obtained with the linear regression between the corrected values after calibration $v_{corr}$ and the true values $v_{true}$ enables determining the improvement brought by the calibration algorithm. This approach with linear regression is used in the literature on calibration of individual instruments [9]. In our case, this error model is perfectly appropriate to evaluate the remaining error after calibration, considering the drift model defined in Section 4.

The case study of Section 4 is analysed here again with the error model. The results associated to all the tables and figures presented are provided. First, the overall results obtained with this approach for the slope and intercept of the linear model, plus the score of the regression, for example, the coefficient of determination of the regression, are in Table 4. Then, results are reported in Table 5 for the same particular instruments considered in Section 4. From these tables, the observations that can be made are consistent with those made with Table 1 and Table 2.

These tables also seem to confirm a bias for the strategy SVR-KF: the standard deviation of the slope and intercept among sensors is small, the mean slope is close to 1 but the mean intercept is equal to 10. However, the poor mean score of the regression invites to be careful.

Table 6 shows the mean and standard deviation of *a*, *b* and of the score computed weekly for a particular instrument. In this case, the means are different from those of Table 5 for $s_{1}$ (a linear regression is not a linear operation). We observe that the slope is varying significantly for SVR-KF (σ=0.28) but also for AB-(DT, KF) and K-KF (σ=0.28, 0.30 and 0.32 respectively). However, SVR-KF is the only strategy with a mean intercept equal to zero but with a standard deviation of 17. This is confirmed with the curves representing the evolution of the slope, intercept and score of the linear regression in Figure 8. From these figures we observe that both the slope and intercept are locally poor for SVR-KF, although this is not visible when computing the error model on the entire time interval of study. Regarding the other strategies, the intercept and score are quite constant. The slope is also quite constant for SM-(SVD, LS, TLS) but it mainly follows the evolution the slope of the measured valued for AB-(DT, KF) and K-KF. Therefore, the observation made with Table 5 about SVR-KF is not valid locally and can explain why the mean score is poor compared to the ones obtained with the other strategies. Compared to the conclusions of Section 4 based on Figure 7 for SVR-KF (“results with MAE for SVR-KF are nearly always worse than those for AB-DT, AB-KF and K-KF until week 24 but are better afterwards”), we observe the results are generally quite poor, both before and after week 24.

In conclusion, the use of an error model allows generally making the same conclusions as in Section 4. However, clearer information can be derived such as the remaining error after calibration, expressed in terms of gain and offset.

### 5.3. Means of Visualization

Choosing other ways of visualizing results may help to better understand what happens to each instrument. Various representations (Figure 9, Figure 10 and Figure 11) are discussed in terms of suitability for the comparison of methods.

Figure 9 is a boxplot-based representation of the values of MAE computed on the entire time interval of study. A boxplot is provided for each strategy. This representation is a graphical illustration of the information displayed in Table 1 for MAE. In addition, it provides supplementary information compared to the table: quartiles, minimum and maximal values. However, this representation does not show exactly the results for each instrument. One graph is needed for each metric. The plot can also show statistics on the evolution of a metric for a particular instrument like Table 3 but again it can only display information about one metric.

To display values regarding each instrument of a network, a matrix of values can be a solution. A colour scale can make the values more easily readable. Figure 10 is a matrix of the mean of MAE on the entire time interval of study. This figure shows exactly the information for each instrument. The colour scale helps to identify problematic instruments. It is appropriate for real time display. Nevertheless, if multiple metrics (or variables if an error model is considered) are used, one matrix must be plotted for each one. The study of multiple calibration strategies is not possible on a single plot.

To overcome the challenge of displaying information for multiple instrument on a single chart, we propose to use target plots. They enable locating a point along multiple axis (two for a 2D diagram, three for a 3D diagram). Boundaries within which the point should be ideally can be depicted. Each axis can be used for a different metrics. A colour scale can be used to depict a third one. In the previous section, we proposed to use an error model with two variables. Therefore, a target plot is perfectly adapted. Figure 11 is a target plot of the parameters of the error model, each point’s coordinates being the values of the slope and the intercept of the error model for each instrument, computed on the entire time interval of study. Indeed, this diagram shows exactly the information for each instrument. Moreover, statistics like standard deviation can be added with error bars. The target diagram is also appropriate for real time display and results for multiple calibration strategies can be plotted. Thus, it is a very powerful representation, although it is limited to two or three variables, respectively for a 2D plot and a 3D plot (eventually three or four with a colour scale on the markers) to allow a graphical representation, and only one statistic per axis can be added with error bars.

### 5.4. Conclusion

In this section, we investigated further issues regarding the evaluation of performances of in situ calibration strategies in terms of metrics and display. The conclusions are as follows:In the case of an evaluation based on usual metrics, the study with multiple metrics is strongly recommended, with a careful regard on the purpose of each metric. Possible interdependence and underdetermination between them (Note that the set of metrics that is used in this work (or a subset) is not optimal itself regarding these questions. See Reference [40]) may mislead interpretations. We recommend using an error model-based approach, which allows to better capture the correction brought by a calibration algorithm, for instance in terms of gain and offset.The way metrics (or the parameters when an error model is used) are computed (time range on which the calculation is performed, periodicity of the computation) should be consistent with the details of the case study.Computing statistics of metrics over the whole network is advantageous for large sensor networks, but may hide important information regarding to individual sensor (for instance, a specific sensor degrading the overall performances of a given method due to outliers).Graphical visualizations of metrics (or parameters of an error model) can enable a better analysis, notably with a target plot of the model’s parameters when an error model with two or three parameters is used.

## 6. Sensitivity of the Calibration Algorithms to the Specificities of the Case Study

In this section, we discuss the influence of the solutions chosen for the implementation of each step of the framework on the performances of the algorithms studied. First the function used to model the concentration of pollutant is modified. Then, the number of instruments is changed. Afterwards, the influence of the drift model is investigated. Finally, the impacts of a change of parameters for the calibration algorithms is studied. For comparison purposes, the metrics chosen in Section 4 are used, completed with the error model and the target plot proposed in Section 5 when appropriate.

### 6.1. Using a More Realistic Model of the True Values

The model used in Section 4 is simplistic and can legitimately raise questions on the performances of each calibration strategy with a more representative model. In this section, consider the Gaussian Plume model [42,43], a refined model with meteorological parameters, taken from a real-world data set. Otherwise, the space of study and its discretization, the pollutant source location and duration of study, the sensor network, the sensor drift model, and the parameters of the calibration strategies are kept similar to the previous case study.

We suppose that at each time step, the pollutant dispersion to be in steady state. Concentration *C* at instant *t* and position (x,y,z) for a wind direction following *x* is expressed as:$$C\left(x,y,z,t\right)=\frac{Q}{4\pi V_{w}\left(t\right)\sigma_{y}\sigma_{z}}e^{-\frac{y^{2}}{4\sigma_{y}^{2}}}\left(e^{-\frac{{\left(z-H\right)}^{2}}{4\sigma_{z}^{2}}}+e^{-\frac{{\left(z+H\right)}^{2}}{4\sigma_{z}^{2}},}\right)$$
with
$\sigma_{y}$ and $\sigma_{z}$: respectively horizontal and vertical dispersion coefficients*Q*: Emission rate at the source$V_{w}$: Wind speed*H*: Pollutant effective release. $H=h_{s}+\Delta h\left(t\right)$ where
$$\Delta h\left(t\right)=\frac{1.6F^{\frac{1}{3}}x^{\frac{2}{3}}}{V_{w}}\;\mathrm{with}\;F=\frac{g}{\pi }D\left(\frac{T_{s}-T\left(t\right)}{T_{s}}\right)$$-$h_{s}$: pollutant source height-*g*: gravity constant-*D*: volumetric flow-$T_{s}$: source temperature-*T*: ambient temperature

To allow for wind direction changes, the cylindrical coordinate system is used to rotate the plume.

Emission rate, temperature of the source and dispersion coefficients are supposed constant over the simulation range. Ambient temperature, wind speed and wind direction are extracted from METAR weather reports collected over a year. An example of pollution map obtained for a given altitude *z* is represented in Figure 12.

Figure 13 is a temporal plot for a particular instrument like in Section 4. From this plot we observe that the true signal appears to vary more quickly from high to low values. This is due to the fact that only one source of pollutant is considered in the area, and that the wind varies quickly from one direction to another in this dataset. Moreover, as we assumed that the concentration at each time step could be represented following the equation of a stationary Gaussian Plume, the pollutant quickly spreads at some points if the wind direction changes a lot. In practice, the pollutant may not disperse as fast.

Results with usual metrics and statistics on the parameters of the linear error model are presented in Table 7 on the entire network. From this table, we observe first that no calibration strategy provides less error than without calibration, even for SM-(SVD, LS, TLS), the strategies that had the best performances in Section 4. This is also the case in Table 8 for the same particular instruments considered in Section 4, despite apparently satisfying results according to MAE but not according to the slope and intercept notably. The results for each instrument are depicted in detail in Figure 14 with the target plot of the parameters of the error model for all the instruments.

Using a more realistic model of dispersion obviously produces a more complex signal that may require a higher density of sensors to capture the phenomena. Indeed, the subspace size for SM-(SVD, LS, TLS) is equal to 11 (it was 2 in Section 4.2.1). This is still fitting the oversampling condition required for these algorithms, but it could explain the actual results: there may be not enough measuring instruments to capture all the features of the measurand. The same reasoning applies for SVR-KF which is also model based. For the other strategies, for which the prediction of true values is based on neighbouring nodes, the quality of the results which were already poor in Section 4 is even worse.

In conclusion, the influence of the model used to derive the true values of the measuring instruments is important. Considering cases on urban air pollution monitoring, the dispersion of pollutant for cities with different street layout may lead to different concentration fields despite identical pollutant sources and experimental conditions. Therefore, even with an identical sensor network with instruments drifting in the same way, in situ calibration strategies could produce corrections of a variable quality.

### 6.2. Density of the Sensor Network

The density of the network of sensors is crucial to capture fine spatial features of the phenomena. In this section, we investigate the impact on the performances of the calibration strategies.

First, we replay the initial case study (2D Gauss dispersion model) with regular grid networks with one sensor every $\frac{1000}{n}$
m, n∈[2,...,10]. The size of the network, $\left|S\right|=n^{2}$ grows from 4 to 100. It was equal to 16 Section 4. All other parameters are kept.

Evolution of the mean of the MAE of the network with |S| is showed in Figure 15. From it, we observe that increasing the number of instruments improves the results with strategies K-KF and SVR-KF. However, the results are not significantly impacted by |S| for strategies AB-(DT, KF) and SM-(SVD, LS, TLS) compared to the baseline case (|S|=16). It can be explained by the simplicity of the model used to derive the true values. Moreover, for these strategies, slightly better results are obtained with a smaller number of instruments than with |S|=16. It seems that such a number (4 or 9 nodes) already enables capturing enough information to calibrate instruments for SM-(SVD, LS, TLS). For AB-(DT, KF), the prediction based on averaging values of neighbour sensors cannot be improved by increasing the number of instruments. Indeed, the network is blind, and sensors are all drifting at the same time, following the same model, with different values of drift. For SVR-KF, issues identified in Section 4 and Section 5 seem to prevail despite an improvement of the performances with more instruments (but not with a lower number of nodes). However, in a general manner for K-KF, the more instruments are deployed, the more the kriging seems to provide accurate predictions. Results have still to be improved for |S|=100 though.

Thus, the number of instruments has an influence, though limited, on the results of the in situ calibration strategies. We also conjecture that the way instruments are deployed can have an influence, as it is crucial for an efficient reconstruction of the signal [44].

We also replay Gaussian Plume case study with a regular grid network of 100 sensors. We conjectured in Section 6.1 that increasing the density of instruments could improve on the poor calibration performances witnessed. Unexpectedly, our results contradict this intuition: the increased number of sensors does not improve the performances of the considered strategies.

Results with usual metrics and statistics on the parameters of the linear error model are presented in Table 9 for the entire network. From this table, we indeed observe that no calibration strategy allows to obtain less error than without calibration. In Table 10 results are quite equivalent to the previous ones in orders of magnitude. SM-(SVD, LS, TLS) strategies have abnormal performances, notably due to outliers in the results for some instruments. Figure 16 depicts the target plot of the parameters of the error model for all the instruments.

For the strategies AB-DT, AB-KF and K-KF, which perform the prediction of true values based on neighbouring nodes, the fact that instruments are drifting altogether may have a more important effect than the density of the network. For SM-(SVD, LS, TLS), the subspace of the signal, which is at heart of these algorithms, was chosen considering the same criteria on the sum of the explained variance ratios. In this case, 13 components of the PCA with the highest variance ratios were kept (11 in Section 6.1). Thus, setting that a subspace is satisfying if the sum of the explained variance ratios of its components is greater than a threshold may be not sufficient to define it properly. The same reasoning applies for SVR-KF which is also model based.

### 6.3. Sensor Modelling

The modelling of the sensors used combines a simple drift model and the assumption that there are no other faults occurring to sensors than drifting. This hypothesis is not realistic, in particular when considering low-cost sensors. In this section, we introduce a more realistic drift model and evaluate the robustness of calibration strategies when the measurements are corrupted by other faults.

The model for the simulation of the true values, the sensor network and the parameters of the calibration strategies used in Section 4 are considered.

#### 6.3.1. Drift Model

To determine the influence of the drift complexity, two models are considered, with $t_{\mathrm{start}\;\mathrm{drift}}$ being the instant when the instrument begins to drift:Random gain and offset increase (RGOI): Gain $G(s_{i},t)$ and offset $O(s_{i},t)$ drift of sensor $s_{i}$ are computed each time step following
$$G\left(s_{i},t\right)=\left\{\begin{array}{cc} 1 & \mathrm{if}\;t<t_{\mathrm{start}\;\mathrm{drift}}\\ G\left(s_{i},t-1\right)+\delta G\left(s_{i},t\right) & \mathrm{if}\;t\geq t_{\mathrm{start}\;\mathrm{drift}} \end{array}\right.$$
$$\;\mathrm{with}\;\forall t,\delta G\left(s_{i},t\right)\;\mathrm{drawn}\;\mathrm{following}\;U\left(0,\delta G_{max}\right)$$
$$O\left(s_{i},t\right)=\left\{\begin{array}{cc} 0 & \mathrm{if}\;t<t_{\mathrm{start}\;\mathrm{drift}}\\ O\left(s_{i},t-1\right)+\delta O\left(s_{i},t\right) & \mathrm{if}\;t\geq t_{\mathrm{start}\;\mathrm{drift}} \end{array}\right.$$
$$\;\mathrm{with}\;\forall t,\delta O\left(s_{i},t\right)\;\mathrm{drawn}\;\mathrm{following}\;U\left(0,\delta O_{max}\right)$$
with $\delta G_{max}$ and $\delta O_{max}$ being respectively the maximal gain and offset possible increase per time step.Continuous gain and offset increase (CGOI): A constant gain $\delta G(s_{i})$ and offset $\delta O(s_{i})$ increase of sensor $s_{i}$ are drawn from uniform laws at the beginning of the drift and are added to respectively to the gain and offset of the instrument at each time step.
$$G\left(s_{i},t\right)=\left\{\begin{array}{cc} 1 & \mathrm{if}\;t<t_{\mathrm{start}\;\mathrm{drift}}\\ G\left(s_{i},t-1\right)+\delta G\left(s_{i}\right) & \mathrm{if}\;t\geq t_{\mathrm{start}\;\mathrm{drift}} \end{array}\right.$$
$$\;\mathrm{with}\;\delta G\left(s_{i}\right)\;\mathrm{drawn}\;\mathrm{following}\;U\left(0,\delta G_{max}\right)$$
$$O\left(s_{i},t\right)=\left\{\begin{array}{cc} 0 & \mathrm{if}\;t<t_{\mathrm{start}\;\mathrm{drift}}\\ O\left(s_{i},t-1\right)+\delta O\left(s_{i}\right) & \mathrm{if}\;t\geq t_{\mathrm{start}\;\mathrm{drift}}] \end{array}\right.$$
$$\;\mathrm{with}\;\delta O\left(s_{i}\right)\;\mathrm{drawn}\;\mathrm{following}\;U\left(0,\delta O_{max}\right)$$
with $\delta G_{max}$ and $\delta O_{max}$ being respectively the maximal gain and offset possible increase per time step.

For both models, measured values are expressed following:$$v\left(s_{i},t\right)=G\left(s_{i},t\right)\;\cdot \;v_{true}\left(s_{i},t\right)+O\left(s_{i},t\right)$$

In this case, $\delta G_{max}=6\times 10^{-5}$, and $\delta O_{max}=0.03$.

There is an offset in RGOI and CGOI drift models. The estimation of the offset could not be carried out with SM-(SVD, LS, TLS) because the phenomena has not a null average on the give time span of calibration. Hence, the offsets are expected to be wrong for these strategies.

Results with usual metrics and statistics on the parameters of the linear error model are presented in Table 11 and Table 12. From these tables and compared to Table 1 and Table 4, we observe that very different results can be obtained depending on the drift model and notably for SM-(SVD, LS, TLS). For these strategies, the measured values with RGOI or CGOI drift models are less improved by the calibration strategies even if we take into account that we expected to have incorrect offsets. Indeed, the average slope is worse than the one in Table 4, with an equivalent standard deviation. For both drift models, SVR-KF seems to improve the measured values according to MAE, MAPE, RMSE and the Pearson correlation coefficient but not according to the error model with a poor slope (0.76) and a high intercept (20) in average. AB-(DT,KF) and K-KF give corrected values equivalent to the measured values with the RGOI model. They are slightly improved if the measured values are built with the CGOI model and more notably for K-KF.

Thus, the drift model used to derive the measured values can drastically change the corrected values obtained after calibration depending on the algorithm used.

#### 6.3.2. Other Faults

As stated in Section 3, sensors are subject to other faults but drift. Even if calibration algorithms are not capturing these faults, the quality of input data impacts the performances of calibration. In Reference [45], it was shown that the outliers contained in the time series of measured values have a more important influence on the calibration results than the average quality of the data. As an early work on the topic, Ramanathan et al. [46] observed that the measured values were sometimes corrupted, even with an individual field calibration for each sensor performed with a mobile chemistry laboratory at the time of deployment. In this section, we investigate the influence of other faults on the calibration results.

Let us assume that in addition to a WGLI drift, all the sensors are undergoing after $t_{\mathrm{start}\;\mathrm{drift}}$:
**Noise** Disturbances that may be correlated or not to the signal are added to the true values. We model it as a random variable following a normal law. For each instrument $s_{i}$, the value of the noise $\varepsilon (s_{i},t)$ is drawn from $N(0,\varepsilon_{max})$ at each time step, with $\varepsilon_{max}=20$ here.**Spikes** Values that are very different from the ones expected with no physical relationship with the measurand. In this study, a spike occurs depending on the random variable $p_{\psi }\left(s_{i},t\right)$ that follows U(0,1). If $p_{\psi }\left(s_{i},t\right)<0.05$, a spike is added to the measured value $v(s_{i},t)$. The value of the spike is equal to $\psi \;\cdot \;v(s_{i},t)$, with $\psi (s_{i},t)$ following U(−1,1).

Thus, the measured values of an instrument $s_{i}$ are equal to:$$v\left(s_{i},t\right)=\left\{\begin{array}{cc} G\left(s_{i},t\right)\;\cdot \;v_{true}\left(s_{i},t\right) & \mathrm{if}\;t<t_{\mathrm{start}\;\mathrm{drift}}\\ G\left(s_{i},t\right)\;\cdot \;v_{true}\left(s_{i},t\right)+\varepsilon \left(s_{i},t\right) & \mathrm{if}\;t\geq t_{\mathrm{start}\;\mathrm{drift}}\;\mathrm{and}\;p_{\psi }\left(s_{i},t\right)\geq 0.05\\ \left(G\left(s_{i},t\right)+\psi \left(s_{i},t\right)\right)\;\cdot \;v_{true}\left(s_{i},t\right)+\varepsilon \left(s_{i},t\right) & \mathrm{if}\;t\geq t_{\mathrm{start}\;\mathrm{drift}}\;\mathrm{and}\;p_{\psi }\left(s_{i},t\right)<0.05. \end{array}\right.$$

Results with usual metrics and statistics on the parameters of the linear error model are presented in Table 13. While SM-(SVD, LS, TLS) strategies worked very well for instruments suffering from drift only, their performances degrade significantly after introducing these faults in the measured values. For the other strategies, noise and spikes do not seem to degrade the results although the performances were fairly unsatisfying from the start like in Section 4. It is expected since this kind of fault is not considered by the calibration strategies. It is henceforth less a matter of performance than an issue of robustness of each method against abnormal values. In practice, one should try to identify and correct as many faults as possible, for example, by filtering spikes and averaging noise, before applying an in situ calibration algorithm.

### 6.4. Parameters of Calibration Strategies

Finally, the last step that can influence the results is the configuration of the calibration strategies. Indeed, the strategies considered in Section 4 have multiple parameters.

Multiple studies can be performed for each parameter of each strategy we considered. As an example, we only investigate the case of SM-(SVD, LS, TLS). The influence of the frequency of computation and of the duration of the time range used to select the measured values used is studied. Note that this aspect was not treated in the original publication [18]. We consider here that the period of computation and the duration of the time range for the selection of the measured values are equal and denoted *w*. The same environment, sensor network and drift model as in Section 4 are considered.

Figure 17 represents the evolution of the mean of MAE over the network for the considered strategies as a function of *w*. The best result is obtained with w=7, which was our original value in Section 4, considering the average of the error for the three strategies with the same *w*. Nevertheless, changing the value of *w* may lower the improvement brought by the in situ calibration for low and high *w* but the corrected values are still closer to the true values compared to the measured values.

We also changed the criteria to determine the signal subspace. The sum of the explained variance ratio of the components obtained by PCA must be greater $1-1\times 10^{-30}$ instead of $1-1\times 10^{-3}$ previously. This resulted in a change of the subspace with four components considered instead of two. The same study with varying *w* was conducted and results are shown in Figure 18.

We still observe that the best results are obtained for w=7: for most of the values of *w*, the algorithms degrade the quality of the measurements. This is even more significant for SM-SVD and SM-TLS. We however do not claim that w=7 is an optimal parameter since we only showcase this feature of the framework: it enables a full parametric optimization of the methods. Calibration methods are indeed very sensitive to the tuning of several parameters and a comparative study should include a parametric optimization.

### 6.5. Summary of the Results

In this section, we changed independently several parameters: the model of the phenomenon, the number of instruments, the faults introduced, notable the drift model, and the parameters of the calibration strategies for SM-(SVD, LS, TLS) that gave the best results previously.

The results obtained are reported in Figure 19. From a general perspective, we observed that:increasing the complexity of the model of the phenomenon drastically worsen the results for all the strategieschanging the number of instruments between |S|=4 and |S|=100 in our case does not have a determining effect.the model of the drift strongly impacts the resultsthe presence of other faults in the measured values can degrade the resultsthe (correct) adjustment of parameters of calibration strategies may have a significant effect

Therefore, comparing in situ calibration strategies requires multiple precautions on how the corrected values were obtained and on what the measured values were based to provide fair and correct conclusions.

## 7. Discussion and Conclusions

In this paper, we have presented the first framework for the simulation of sensor networks that enables a systematic comparison of in situ calibration strategies with reproducibility, and scalability. We have showcased this methodology by comparing several calibration strategies for blind and static sensor networks. Our results provide several engineering insights on the design of a sensor network.

Based on the methodology described in Section 3, we have proposed a case study in Section 4 concerning blind and static sensor networks. Although we have shown that several strategies have better performances than the others, these results have to be tempered as the conclusions depends on how the evaluation is conducted. We have proposed solutions to conduct a balanced assessment of the performances in terms of metrics used in Section 5. It is still challenging for sensor networks with a high number of instruments that require multiple scale analysis. In addition, we have shown in Section 6 that results may not be the same depending on choices made during the design of the case study regardless on how the evaluation is carried out. We have developed variations of our initial case study by changing the model computing the true values of the measurand, the number of instruments, the fault model of the sensors and parameters of calibration strategies. The detailed data and codes that were used in this work are provided as Appendix A.

Our results highlight the dependence of the performances of in situ calibration algorithms to the case study. Besides, these performances seem to be often limited, even on the relatively simple environmental models discussed here. The latter explains why more complex and accurate environmental models were not tested here. Finding the good trade-off between simplicity and realism of the simulation models is important. At first sight, the goal could be to consider models as realistic as possible everywhere. However, the accuracy of simulation-based environmental models remains challenging to establish, and costly to run, while simplified models such as those used in the paper may be sufficient when the goal is to compare methods to single out the most promising ones among several for a predefined case study. This is also true for the models to simulate the measuring chain and the deployment of the instruments.

It allows to point out an additional advantage to the framework proposed in this paper: its suitability as a design tool for sensor networks. For instance, after having determined the number of instruments (and their positions) required to cover a given area with a static sensor network, the best in situ calibration strategy and its parameters could be identified among a set of algorithms applicable to this particular network. In such cases however, a stronger focus on the accuracy of the environmental model should be put whenever possible. Beyond the quality of the models, the tuning of the strategies is crucial for an objective evaluation. The reproducibility provided by the framework is an asset to conduct a thorough parametric optimization and compare the best results of each calibration.

To conclude, we can point out the following engineering insights on the design of a sensor network and the evaluation of in situ calibration strategies.

There is actually no method that is universally outperforming the other. The best calibration strategy to apply depends on the deployment of the sensors and the exploitation of the measurements.Even on the same scenario, two strategies can outperform each other depending on which metrics of performance is considered and the way they are computed. The metrics and their means of visualization have to be chosen accordingly to the target exploitation of the sensor network, to focus on the relevant features. We also advise to use an error model in order to get meaningful details on the performances.Increasing the density of the deployment of sensors does not always lead to better performances of a calibration strategy.In some case, the quality of data can be degraded by a calibration strategy. Besides, other faults than drift will happen, in particular when considering low-cost sensors with low-quality electronics. To cope with robustness issues of the calibration strategy, it seems relevant to process these errors upstream.

## Figures and Tables

**Figure 1 sensors-20-04577-f001:**
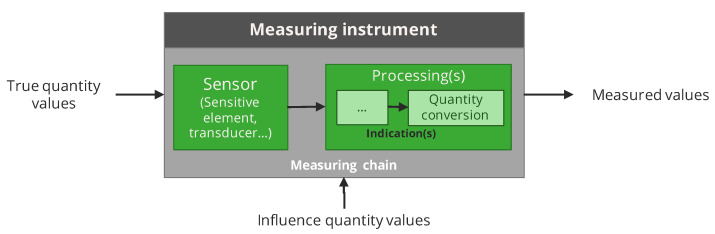
Model adopted for the measuring instruments. It is considered as a grey box, for example, an algorithm mimics the behaviour of the components of the measuring chain without however modelling each part individually.

**Figure 2 sensors-20-04577-f002:**
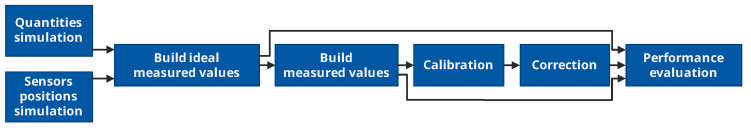
Schematic diagram of the methodology proposed for in situ calibration strategies evaluation.

**Figure 3 sensors-20-04577-f003:**
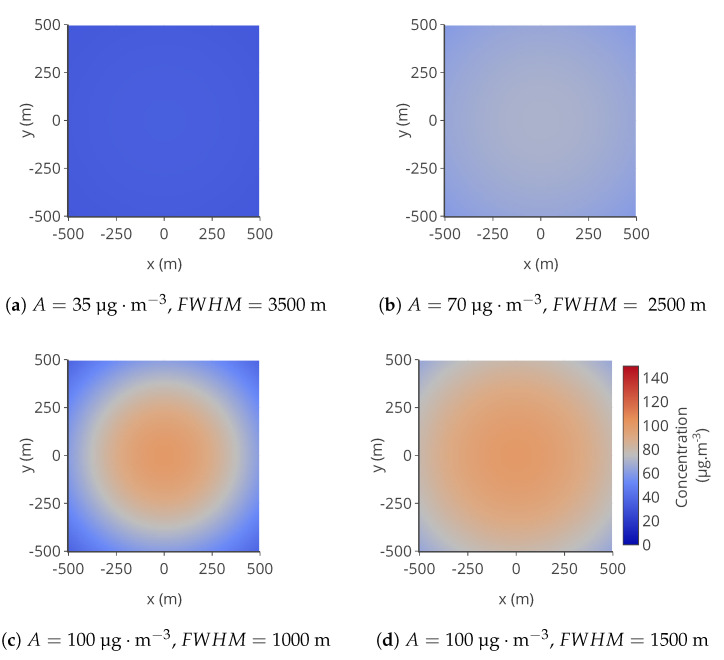
Examples of maps of concentration *C* used following $C\left(x,y\right)=A\exp\left(-{\left(2\ln\left(2\right)\right)}^{2}\frac{x^{2}+y^{2}}{FWHM^{2}}\right)$ for given *A* and FWHM.

**Figure 4 sensors-20-04577-f004:**
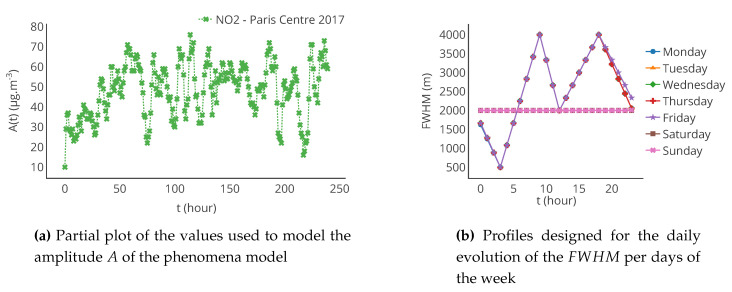
Evolution of *A* and FWHM for the modelling of the concentration of pollutant.

**Figure 5 sensors-20-04577-f005:**
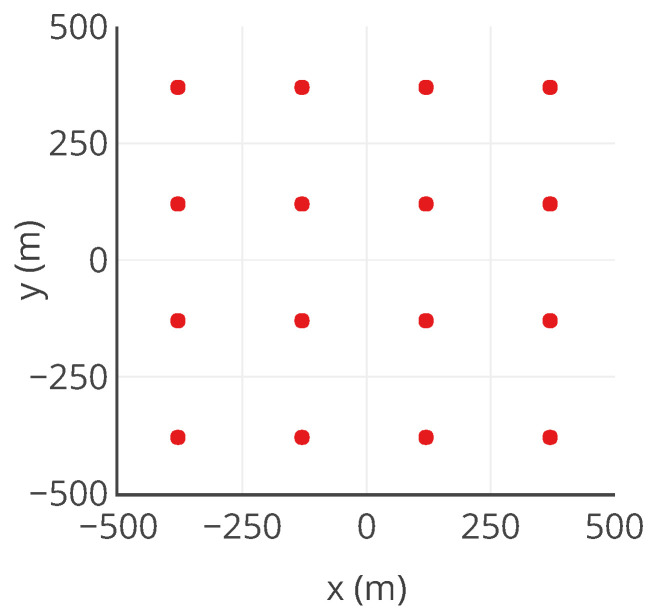
Positions of the 16 sensors considered in the case study, deployed uniformly in the field.

**Figure 6 sensors-20-04577-f006:**
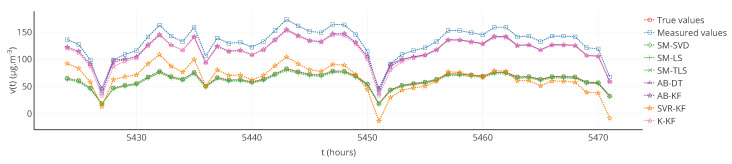
True values, measured values and corrected values with the strategies considered for a particular sensor $s_{1}=2$ between t=5424h and t=5471h. SM-(SVD, LS, TLS) and SVR-KF seem to provide better results than AB-DT, AB-KF and K-KF.

**Figure 7 sensors-20-04577-f007:**
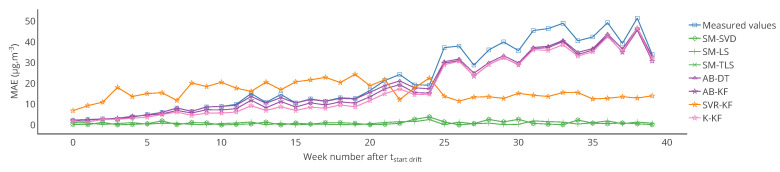
Evolution of the MAE computed for each week of the drift period between the drifted values and the true values, and between the corrected values for each strategy and the true values for a particular sensor, after the start of drift. MAE for SVR-KF is nearly always worse than those for AB-DT, AB-KF and K-KF until week 24 but are better afterwards. The performances of SVR-KF could be explained by the presence of a bias, at least according to this metrics, as its evolution is quite flat.

**Figure 8 sensors-20-04577-f008:**
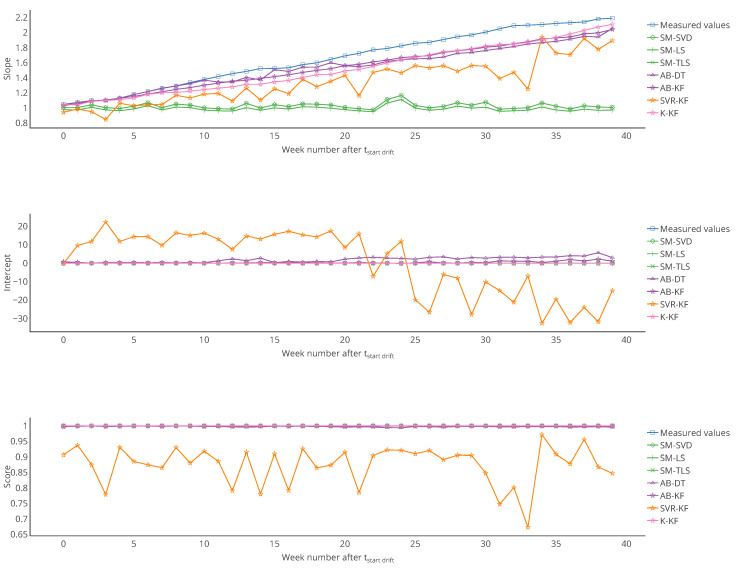
Evolution of the slope, intercept and score of the error model of $s_{1}$, computed on each week of the drift period. From these figures we observe that both the slope and intercept are locally poor for SVR-KF, although this is not visible when computing the error model on the entire time interval of study. This explains the poor values of the score for this algorithm. Regarding the other strategies, the intercept and score are quite constant. The slope is also quite constant for SM-(SVD, LS, TLS) but it mainly follows the evolution the slope of the measured valued for AB-(DT, KF) and K-KF.

**Figure 9 sensors-20-04577-f009:**
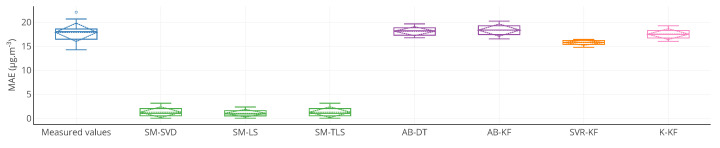
Box plots of the MAE, computed on the entire time interval of drift, of the 16 nodes of the network without calibration and with SM-SVD. It is a graphical representation of the information displayed in Table 1 for MAE. Appendix A is provided compared to the table: quartiles, minimum and maximal values.

**Figure 10 sensors-20-04577-f010:**
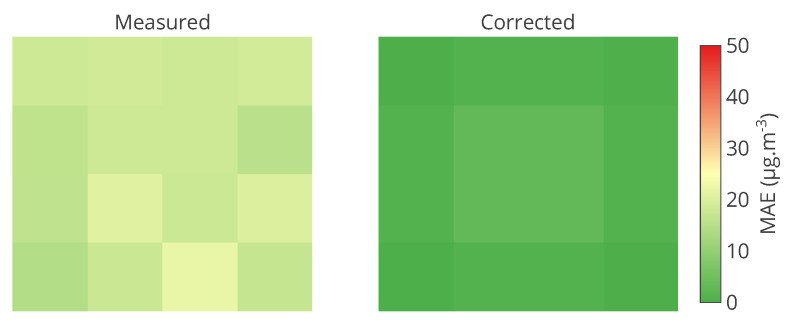
Matrix of the MAE, computed on the entire time interval of drift, of the 16 nodes of the network without calibration and with SM-SVD. It shows exactly the information for each instrument. The colour scale can help to identify problematic instruments.

**Figure 11 sensors-20-04577-f011:**
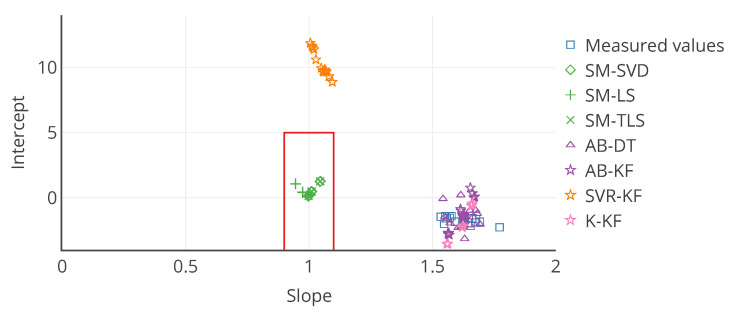
Target plot of the 16 nodes as a function of their slope and intercept in the error model for each calibration strategy. The slope and intercept are computed on the entire time interval of drift. In this case, it allows to locate an instrument according to the slope and intercept of its associated error model. The red rectangle is an example of area that can be defined to quickly identify instruments that are not satisfying a requirement, here a slope in [0.9;1.1] and an intercept in [−5;5].

**Figure 12 sensors-20-04577-f012:**
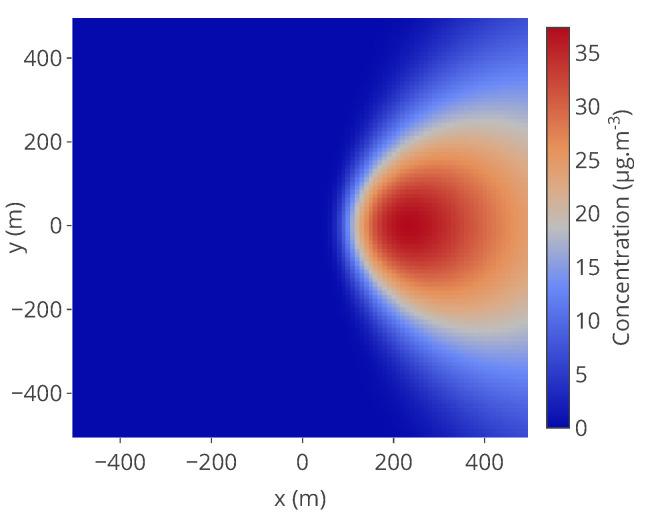
Example of concentration map used and based on the Gaussian plume model. The pollutant source is at coordinates $\left(x_{s},y_{s}\right)=\left(0,0\right)$, $h_{s}=25\;m$, the map is observed at z=2m. The other parameters are: {*T* = 25 °C, *V_w_* = 10 m s^−1^, *T_s_* = 30 °C, *g* = 9.8 m s^−2^, *D* = 1.9 × 10^−9^ m^3^ s^−1^, *Q* = 5 × 10^−3^ kg s^−1^, *σ_y_* = 1.36|*x*−*x_s_*|^0.82^, *σ_z_* = 0.275|*x*−*x_s_*|^0.69^}. Wind direction is equal to 0° here.

**Figure 13 sensors-20-04577-f013:**
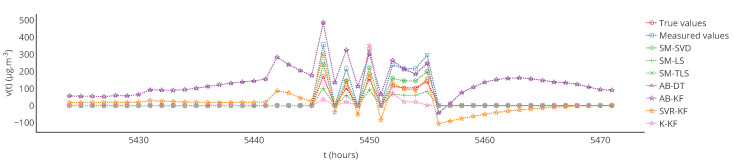
True values, measured values and corrected values with the strategies considered for a particular sensor between t=5424h and t=5471h with the Gaussian plume model. Note that the profiles of the curves are very different from those of Figure 6 as instruments are not necessarily exposed to the plume of pollutant due to the wind direction. In this case it is exposed to the pollutant between t=5445h and t=5456h.

**Figure 14 sensors-20-04577-f014:**
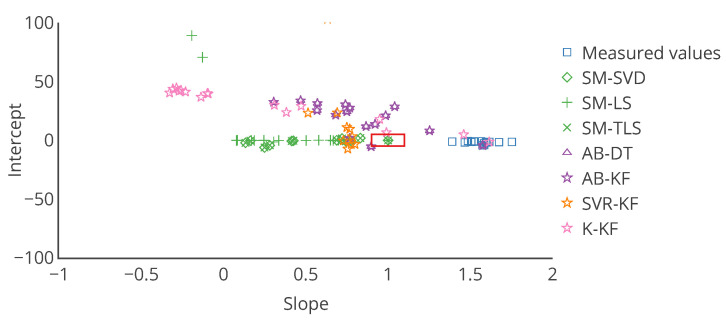
Target plot of the 16 nodes as a function of their slope and intercept in the error model, computed on the entire time interval of drift, for each calibration strategy and with the Gaussian plume model. The instruments are not all present, notably for those after the correction by SVR-KF. Axis were truncated to keep the plot readable. The red rectangle depicts the same requirement as in Figure 11 for example, a slope in [0.9;1.1] and an intercept in [−5;5]. Few instruments are inside this area whatever the in situ calibration strategy considered.

**Figure 15 sensors-20-04577-f015:**
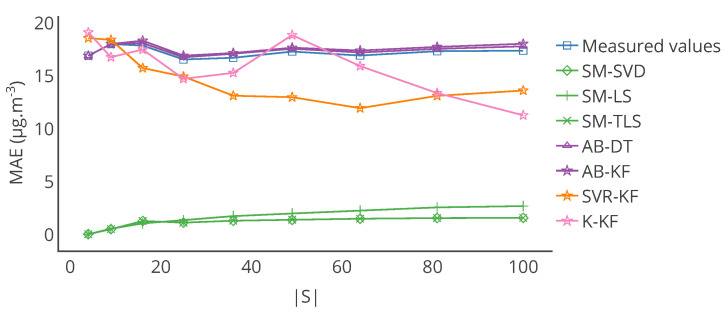
Evolution of the mean of the MAE, computed on the entire time interval of study, of all the nodes of the network, as a function of the number of nodes |S|, for the 2D Gauss model.

**Figure 16 sensors-20-04577-f016:**
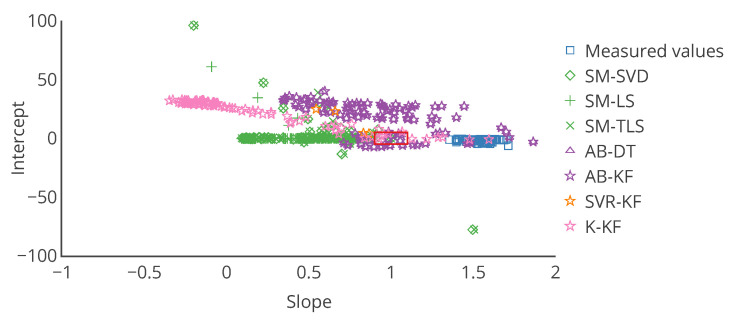
Target plot of the 16 nodes as a function of their slope and intercept in the error model, computed on the entire time interval of study, for each calibration strategy and with the Gaussian plume model. The instruments are not all present: axes were truncated to keep the plot readable. The red rectangle depicts the same requirement as in Figure 11 and Figure 14 for example, a slope in [0.9;1.1] and an intercept in [−5;5]. More instruments are inside this area compared to Figure 14 but most of them are outside whatever the in situ calibration strategy considered.

**Figure 17 sensors-20-04577-f017:**
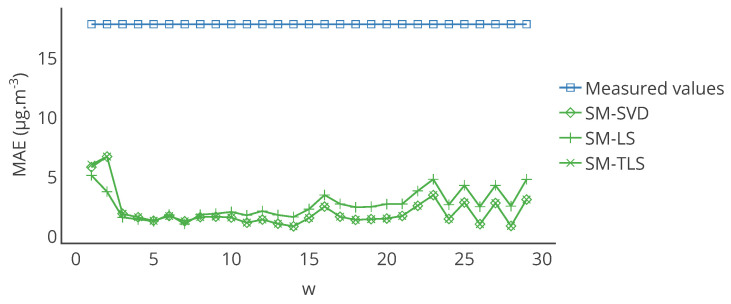
Evolution of the mean of MAE, computed on the entire time interval of study, over the network for the strategies SM-SVD, SM-LS and SM-TLS as a function of *w*. The best result is obtained for w=7, which was our original value in Section 4, considering the average error of the three strategies for the same *w*.

**Figure 18 sensors-20-04577-f018:**
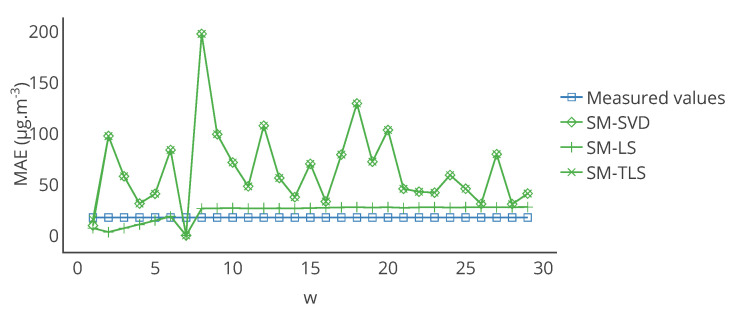
Evolution of the mean of MAE, computed on the entire time interval of study, over the network for the strategies SM-SVD, SM-LS and SM-TLS as a function of *w* with a different signal subspace than in Figure 17. The best results are still obtained for w=7 but for most the values of *w*, the algorithms degrade the quality of the measurements and more significantly for SM-SVD and SM-TLS.

**Figure 19 sensors-20-04577-f019:**
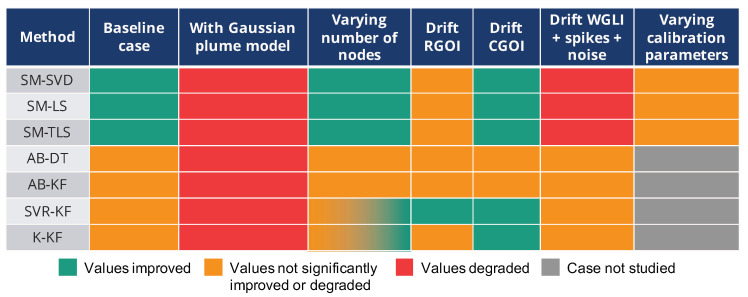
Summary of the results observed by changing the parameters of the baseline case study which was built with a 2D Gaussian function to model the measurand, with the WGLI drift model applied to the measured values of the instruments, with 16 nodes in the network and the values indicated in Section 4 for the parameters of the in situ calibration algorithms. The improvements are judged relatively to results of the baseline case, recalled in the first column. Thus, this table does not indicate that the corrected values are very close to the true values when the measured values are improved.

**Table 1 sensors-20-04577-t001:** Mean and standard deviation of each metric, computed on the entire time interval of drift, over the 16 nodes of the network. SM-(SVD, LS, TLS) strategies have the best results overall whatever the metric considered. SVR-KF provides corrected values only slightly better than before calibration according to MAE and RMSE but not according to MAPE and Pearson correlation coefficient. AB-(DT, KF) and K-KF do not improve the measured values significantly.

	MAE			MAPE			RMSE			Pearson	
	μ	σ		μ	σ		μ	σ		μ	σ
**No calibration**	18	2		53	6		25	2		0.935	0.007
**SM-SVD**	1	1		5	4		2	2		0.994	0.006
**SM-LS**	1	1		3	2		2	1		0.995	0.005
**SM-TLS**	1	1		5	4		2	2		0.994	0.006
**AB-DT**	18	1		55	6		25	1		0.932	0.004
**AB-KF**	18	1		58	11		25	1		0.933	0.003
**SVR-KF**	16	1		83	11		18	1		0.825	0.012
**K-KF**	18	1		53	10		24	1		0.927	0.005

**Table 2 sensors-20-04577-t002:** Values of the metrics, computed on the entire time interval of drift, for two particular sensors of the network $s_{1}=2$ and $s_{2}=10$. Strategies AB-(DT, KF) and (SVR, K)-KF are quite equivalent for these two instruments. The improvements are rather small for both sensors. For SM-(SVD, LS, TLS), results are consistent with the observations of Table 1.

	MAE			MAPE			RMSE			Pearson	
	$s_{1}$	$s_{2}$		$s_{1}$	$s_{2}$		$s_{1}$	$s_{2}$		$s_{1}$	$s_{2}$
**No calibration**	22	18		66	51		30	24		0.924	0.942
**SM-SVD**	1	3		4	10		2	5		0.996	0.983
**SM-LS**	1	2		3	7		2	3		0.997	0.986
**SM-TLS**	1	3		4	10		2	5		0.996	0.983
**AB-DT**	20	17		61	49		26	23		0.933	0.932
**AB-KF**	18	17		56	45		25	23		0.935	0.927
**SVR-KF**	16	16		80	71		19	19		0.828	0.805
**K-KF**	17	16		51	43		24	23		0.930	0.918

**Table 3 sensors-20-04577-t003:** Mean and standard deviation of the weekly mean of each metric for $s_{1}$ during the drift period. From the standard deviations of MAE, MAPE and RMSE, the observations made based on Table 1 could be locally false, for example, a strategy is better than others considering a computation of the metrics over the 12 weeks of drift but not always considering a computation of the metrics over each week.

	MAE			MAPE			RMSE			Pearson	
	μ	σ		μ	σ		μ	σ		μ	σ
**No calibration**	22	16		66	36		25	17		1.000	0.000
**SM-SVD**	1	1		3	3		1	1		1.000	0.000
**SM-LS**	1	1		3	2		1	1		1.000	0.000
**SM-TLS**	1	1		3	3		1	1		1.000	0.000
**AB-DT**	20	13		61	32		22	14		0.999	0.001
**AB-KF**	18	13		56	31		21	14		0.999	0.000
**SVR-KF**	16	4		80	39		18	4		0.935	0.034
**K-KF**	17	14		51	32		20	15		1.000	0.000

**Table 4 sensors-20-04577-t004:** Mean (μ) and standard deviation (σ) of the parameters of the error model and the regression score, computed on the entire time interval of drift, over the 16 nodes of the network. The observations that can be made with these values are identical to those of Table 1. This table also seems to confirm that there is a bias for SVR-KF as the standard deviation of the slope and intercept among sensors is small, with mean slope close to 1 but a mean intercept equal to 10. However, the poor mean score of the regression invites to be careful with this statement.

	Slope			Intercept			Score	
	μ	σ		μ	σ		μ	σ
**No calibration**	1.62	0.07		−2	0		0.87	0.01
**SM-SVD**	1.02	0.02		1	0		0.99	0.01
**SM-LS**	0.97	0.02		1	0		0.99	0.01
**SM-TLS**	1.02	0.02		1	0		0.99	0.01
**AB-DT**	1.62	0.05		−2	1		0.87	0.01
**AB-KF**	1.62	0.04		−1	1		0.87	0.01
**SVR-KF**	1.05	0.03		10	1		0.68	0.02
**K-KF**	1.62	0.04		−2	1		0.86	0.01

**Table 5 sensors-20-04577-t005:** Values of the parameters of the error model and the regression score, computed on the entire time interval of drift, for two particular sensors of the network $s_{1}=2$ and $s_{2}=10$. The observations that can be made with these values are identical to those of Table 2.

	Slope			Intercept			Score	
	$s_{1}$	$s_{2}$		$s_{1}$	$s_{2}$		$s_{1}$	$s_{2}$
**No calibration**	1.77	1.58		−2	−1		0.85	0.89
**SM-SVD**	1.01	1.05		0	1		0.99	0.97
**SM-LS**	0.97	0.94		0	1		0.99	0.97
**SM-TLS**	1.01	1.05		0	1		0.99	0.97
**AB-DT**	1.62	1.55		0	−1		0.87	0.87
**AB-KF**	1.63	1.56		−1	−3		0.87	0.86
**SVR-KF**	1.09	1.01		9	12		0.69	0.65
**K-KF**	1.62	1.56		−2	−4		0.87	0.84

**Table 6 sensors-20-04577-t006:** Mean (μ) and standard deviation (σ) of the parameters of the error model and the regression score for $s_{1}$, computed on each week of the drift period. The slope for SVR-KF varies significantly and the mean intercept is equal to zero with a standard deviation of 17. This is not the case for the other strategies. Thus, the observation made with Table 5 on SVR-KF is not valid locally and can explain why the mean score is poor compared to the ones obtained with the other strategies.

	Slope			Intercept			Score	
	μ	σ		μ	σ		μ	σ
**No calibration**	1.66	0.36		0	0		1.00	0.00
**SM-SVD**	1.03	0.04		0	0		1.00	0.00
**SM-LS**	0.99	0.03		0	0		1.00	0.00
**SM-TLS**	1.03	0.04		0	0		1.00	0.00
**AB-DT**	1.53	0.28		2	1		1.00	0.00
**AB-KF**	1.54	0.30		0	1		1.00	0.00
**SVR-KF**	1.35	0.28		0	17		0.88	0.06
**K-KF**	1.51	0.32		0	0		1.00	0.00

**Table 7 sensors-20-04577-t007:** Mean and standard deviation of each metric, computed on the entire time interval of drift, over the 16 nodes of the network with the Gaussian plume model. No calibration strategy allows to obtain less error than without calibration, even for SM-(SVD, LS, TLS) strategies that had the best performances in Section 4.

	MAE			MAPE			RMSE			Pearson			Slope			Intercept			Score	
	μ	σ		μ	σ		μ	σ		μ	σ		μ	σ		μ	σ		μ	σ
**No calibration**	13	6		17	6		38	21		0.98	0.00		1.56	0.08		−2	1		0.97	0.01
**SM-SVD**	$2.30\times 10^{3}$	$6.12\times 10^{3}$		$2.28\times 10^{5}$	$6.13\times 10^{5}$		$1.39\times 10^{4}$	$3.73\times 10^{4}$		0.55	0.43		−21.11	60.40		$2.49\times 10^{3}$	$6.99\times 10^{3}$		0.48	0.42
**SM-LS**	26	24		952	$2.42\times 10^{3}$		93	138		0.85	0.34		0.34	0.32		10	28		0.83	0.33
**SM-TLS**	$2.30\times 10^{3}$	$6.12\times 10^{3}$		$2.28\times 10^{5}$	$6.13\times 10^{5}$		$1.39\times 10^{4}$	$3.73\times 10^{4}$		0.55	0.43		−21.11	60.40		$2.49\times 10^{3}$	$6.99\times 10^{3}$		0.48	0.42
**AB-DT**	36	3		$2.11\times 10^{3}$	529		54	7		0.62	0.19		0.82	0.31		19	13		0.42	0.24
**AB-KF**	36	3		$2.11\times 10^{3}$	529		54	7		0.62	0.19		0.82	0.31		19	13		0.42	0.24
**SVR-KF**	249	357		$1.51\times 10^{4}$	$2.15\times 10^{4}$		270	380		0.54	0.18		0.83	0.19		234	357		0.32	0.19
**K-KF**	47	22		$2.18\times 10^{3}$	$1.31\times 10^{3}$		85	36		0.11	0.43		0.26	0.66		30	15		0.18	0.27

**Table 8 sensors-20-04577-t008:** Values of the metrics, computed on the entire time interval of drift, for two particular sensors of the network $s_{1}=2$ and $s_{2}=10$, with the Gaussian plume model. No calibration strategy allows to obtain less error than without calibration, even for SM-(SVD, LS, TLS) strategies that had the best performances in Section 4, despite apparently satisfying results according to MAE but not according to the slope and intercept notably.

	MAE			MAPE			RMSE			Pearson			Slope			Intercept			Score	
	$s_{1}$	$s_{2}$		$s_{1}$	$s_{2}$		$s_{1}$	$s_{2}$		$s_{1}$	$s_{2}$		$s_{1}$	$s_{2}$		$s_{1}$	$s_{2}$		$s_{1}$	$s_{2}$
**No calibration**	12	12		19	13		36	35		0.98	0.98		1.75	1.47		−1	−2		0.96	0.97
**SM-SVD**	6	33		108	201		24	269		0.83	0.06		0.83	0.25		2	−6		0.68	0.00
**SM-LS**	8	25		55	86		20	57		0.97	0.97		0.58	0.17		0	0		0.94	0.93
**SM-TLS**	6	33		108	201		24	269		0.83	0.06		0.83	0.25		2	−6		0.68	0.00
**AB-DT**	37	32		$2.59\times 10^{3}$	$1.61\times 10^{3}$		51	45		0.57	0.79		0.77	0.92		27	14		0.33	0.63
**AB-KF**	37	32		$2.60\times 10^{3}$	$1.61\times 10^{3}$		51	45		0.57	0.79		0.77	0.92		28	14		0.33	0.63
**SVR-KF**	24	387		$1.31\times 10^{3}$	$2.60\times 10^{4}$		34	415		0.69	0.34		0.78	0.98		3	382		0.48	0.12
**K-KF**	27	63		818	$3.41\times 10^{3}$		55	100		0.60	−0.25		0.95	-0.26		18	42		0.36	0.06

**Table 9 sensors-20-04577-t009:** Mean and standard deviation of each metric, computed on the entire time interval of study, over the 100 nodes of the network with the Gaussian plume model. Again, no calibration strategy allows to obtain less error than without calibration.

	MAE			MAPE			RMSE			Pearson			Slope			Intercept			Score	
	μ	σ		μ	σ		μ	σ		μ	σ		μ	σ		μ	σ		μ	σ
**No calibration**	13	7		17	6		38	25		0.98	0.00		1.55	0.07		-2	1		0.97	0.01
**SM-SVD**	$1.04\times 10^{12}$	$1.04\times 10^{13}$		$1.04\times 10^{14}$	$1.04\times 10^{15}$		$6.32\times 10^{12}$	$6.32\times 10^{13}$		0.55	0.37		$-1.01\times 10^{11}$	$1.01\times 10^{12}$		$1.22\times 10^{12}$	$1.22\times 10^{13}$		0.43	0.38
**SM-LS**	$4.77\times 10^{8}$	$4.77\times 10^{9}$		$4.77\times 10^{10}$	$4.77\times 10^{11}$		$2.91\times 10^{9}$	$2.91\times 10^{10}$		0.91	0.26		$-4.63\times 10^{7}$	$4.63\times 10^{8}$		$5.63\times 10^{8}$	$5.63\times 10^{9}$		0.89	0.26
**SM-TLS**	$8.67\times 10^{11}$	$8.67\times 10^{12}$		$8.67\times 10^{13}$	$8.67\times 10^{14}$		$5.29\times 10^{12}$	$5.29\times 10^{13}$		0.54	0.37		$-8.43\times 10^{10}$	$8.43\times 10^{11}$		$1.02\times 10^{12}$	$1.02\times 10^{13}$		0.42	0.38
**AB-DT**	37	3		$2.09\times 10^{3}$	580		57	8		0.63	0.20		1.02	0.71		17	14		0.43	0.25
**AB-KF**	37	3		$2.10\times 10^{3}$	578		57	8		0.63	0.20		1.02	0.71		17	14		0.43	0.25
**SVR-KF**	412	208		$2.68\times 10^{4}$	$1.33\times 10^{4}$		432	232		0.51	0.18		0.82	0.14		415	208		0.29	0.16
**K-KF**	39	21		$1.66\times 10^{3}$	973		71	41		0.10	0.44		0.19	1.38		23	11		0.20	0.30

**Table 10 sensors-20-04577-t010:** Values of the metrics, computed on the entire time interval of study, for two particular sensors of the network $s_{1}=2$ and $s_{2}=10$, with the Gaussian plume model. For these instruments, the results are quite equivalent to those of Table 8 in orders of magnitude.

	MAE			MAPE			RMSE			Pearson			Slope			Intercept			Score	
	$s_{1}$	$s_{2}$		$s_{1}$	$s_{2}$		$s_{1}$	$s_{2}$		$s_{1}$	$s_{2}$		$s_{1}$	$s_{2}$		$s_{1}$	$s_{2}$		$s_{1}$	$s_{2}$
**No calibration**	10	10		22	22		23	21		0.98	0.99		1.59	1.54		−1	−1		0.97	0.97
**SM-SVD**	5	3		36	25		13	6		0.92	0.99		0.84	0.87		0	0		0.85	0.98
**SM-LS**	6	4		42	30		11	6		0.99	1.00		0.71	0.84		0	0		0.99	0.99
**SM-TLS**	5	3		36	25		13	6		0.92	0.99		0.84	0.87		0	0		0.85	0.98
**AB-DT**	37	37		$2.16\times 10^{3}$	$2.08\times 10^{3}$		53	54		0.59	0.63		1.11	1.28		20	16		0.35	0.39
**AB-KF**	38	37		$2.17\times 10^{3}$	$2.09\times 10^{3}$		53	54		0.59	0.63		1.11	1.28		20	16		0.34	0.39
**SVR-KF**	373	309		$2.22\times 10^{4}$	$1.75\times 10^{4}$		374	311		0.63	0.68		0.89	1.02		375	309		0.39	0.46
**K-KF**	9	10		44	62		18	22		0.95	0.90		1.31	1.29		1	1		0.91	0.81

**Table 11 sensors-20-04577-t011:** Mean and standard deviation of each metric, computed on the entire time interval of study, over the 16 nodes of the network with the RGOI drift model. Results are very different with this model compared to Table 1 and Table 4, notably for SM-(SVD, LS, TLS) even if we take into account that we expected to have incorrect offsets.

	MAE			MAPE			RMSE			Pearson			Slope			Intercept			Score	
	μ	σ		μ	σ		μ	σ		μ	σ		μ	σ		μ	σ		μ	σ
**No calibration**	56	0		273	46		65	0		0.65	0.00		1.41	0.00		43	1		0.42	0.01
**SM-SVD**	50	3		248	32		56	3		0.67	0.01		1.24	0.01		42	2		0.45	0.01
**SM-LS**	46	1		232	43		52	1		0.68	0.00		1.18	0.04		40	0		0.46	0.00
**SM-TLS**	50	3		248	32		56	3		0.67	0.01		1.24	0.01		42	2		0.45	0.01
**AB-DT**	56	0		274	48		65	0		0.65	0.00		1.41	0.01		43	1		0.42	0.00
**AB-KF**	56	1		276	54		65	1		0.64	0.00		1.41	0.03		43	1		0.42	0.00
**SVR-KF**	14	0		86	13		16	1		0.84	0.01		0.76	0.02		20	1		0.71	0.02
**K-KF**	56	1		276	54		65	1		0.64	0.00		1.41	0.03		43	1		0.41	0.00

**Table 12 sensors-20-04577-t012:** Mean and standard deviation of each metric, computed on the entire time interval of study, over the 16 nodes of the network with the CGOI drift model. Results are very different with this model compared to Table 1 and Table 4, notably for SM-(SVD, LS, TLS) even if we take into account that we expected to have incorrect offsets.

	MAE			MAPE			RMSE			Pearson			Slope			Intercept			Score	
	μ	σ		μ	σ		μ	σ		μ	σ		μ	σ		μ	σ		μ	σ
**No calibration**	44	29		210	131		51	34		0.75	0.15		1.34	0.20		33	23		0.58	0.23
**SM-SVD**	33	21		162	99		38	24		0.78	0.14		1.18	0.12		27	17		0.63	0.21
**SM-LS**	26	16		133	89		31	19		0.77	0.13		1.07	0.13		21	14		0.61	0.20
**SM-TLS**	33	21		162	99		38	24		0.78	0.14		1.18	0.12		27	17		0.63	0.21
**AB-DT**	46	1		227	44		54	1		0.70	0.01		1.35	0.03		35	1		0.49	0.01
**AB-KF**	47	1		228	46		54	1		0.70	0.00		1.35	0.03		36	1		0.49	0.00
**SVR-KF**	14	0		85	13		16	0		0.84	0.01		0.76	0.02		20	1		0.71	0.02
**K-KF**	20	1		95	22		23	1		0.89	0.00		1.18	0.02		14	1		0.79	0.01

**Table 13 sensors-20-04577-t013:** Mean and standard deviation of each metric, computed on the entire time interval of study, over the 16 nodes of the network with the WGLI drift model, spikes and noise. The results with SM-(SVD, LS, TLS) are significantly influenced by the spikes and noise. The other strategies do not seem to provide more degraded results due to the noise and spikes although the performances were fairly unsatisfying from the start in Section 4.

	MAE			MAPE			RMSE			Pearson			Slope			Intercept			Score	
	μ	σ		μ	σ		μ	σ		μ	σ		μ	σ		μ	σ		μ	σ
**No calibration**	25	2		106	14		33	2		0.79	0.01		1.62	0.07		–2	0		0.62	0.01
**SM-SVD**	60	50		198	144		82	70		0.23	0.33		0.18	0.69		–13	14		0.16	0.25
**SM-LS**	28	6		93	10		33	7		0.35	0.57		0.15	0.26		0	1		0.43	0.08
**SM-TLS**	60	50		198	144		82	70		0.23	0.33		0.18	0.69		–13	14		0.16	0.25
**AB-DT**	21	1		78	17		28	2		0.88	0.01		1.63	0.04		–2	1		0.77	0.02
**AB-KF**	21	1		78	17		28	2		0.88	0.01		1.63	0.04		–1	1		0.78	0.02
**SVR-KF**	18	0		92	15		22	0		0.75	0.01		1.06	0.03		10	1		0.56	0.01
**K-KF**	25	1		107	18		32	1		0.79	0.00		1.61	0.04		–2	1		0.62	0.01

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
