# Peer review of "Framework for the Simulation of Sensor Networks Aimed at Evaluating In Situ Calibration Algorithms"

_sensors, 2020, doi:10.3390/s20164577_

Round 1

Reviewer 1 Report

The authors aim to provide a reliable comparison method among different in-situ calibration methods. This is an important issue to study, especially for the sensor networks literature. 

The examples used to describe the approach are well-defined and executed. This is one of the strengths of the paper.

The high-level description of the framework in Section 3 is limited, it must be extended. 

The approach is based on simulation. There are various simulators for sensor networks and sensor network calibration. One advantage of a study like the one given in this paper would be also determining the requirements of the simulators for different types of calibration strategies. It would be a good addition to discuss the simulator options and implementation requirements.

Format: There are typos, even in the abstract, authors need to proof-read the whole document. The description of some concepts and examples are wordy, which makes it harder to read. It would be better if the authors try to give the main points in a concise manner.

Reviewer 2 Report

This manuscript is a thorough step-wise experiment using controlled data inputs to mimic sensor deployments in environmental conditions. The results could be used by other researchers in developing custom calibrations of sensor data. The statistics used are standard (which is good) and the methods and assumptions are clear, discussed and reasonable. The manuscript needs to be edited to be sure correct English is used. E.g., in the abstract: “For traditionally instruments, this operation” should read “for traditional instruments, this operation…”; and “We propose is as an evaluation” should be “we propose an evaluation…”.  There are a few big picture items that would make this manuscript more valuable:

  • Separating out conclusions for how to account for drift, noise and (presumably aberrant?) spikes, as discussed in section 6.3.2. These are exactly the three main issues sensor networks have so having more discussion than the final sentences of this section would be useful. It looks like in Figure 21 SVR-KF may be the best for dealing with drift?
  • Multiple standard statistics are used (MAE, MAPE, Pearson, etc), but it’s unclear if this collection is needed or if only a subset was used would the results be the same, i.e., should future work utilize these statistics or would just a few be sufficient?
  • The concluding bullet in section 6.5 (increasing the number of instruments does not have a determining effect) is interesting – perhaps this should be “increasing beyond the 16 instruments does not have a determining effect”. I expect that if we decreased the number of instruments our ability to correct drift would decrease – if viable, could the authors please comment on this, as it would be useful to know whether decreasing to say 8 or 4 instruments dramatically impacts the ability to develop a correction.
  • Figure 19 shows w=7 indeed has the lowest MAE, but fairly similar to w=5 and 10. I may have missed this but was there a day-of-week amplitude in emissions done in the dispersion modeling? If not, could the authors please comment on whether w=7 would be appropriate for a real-world correction where Monday-Friday emissions are different than those on Saturday and Sunday? A statement along these lines could be made to make sure future researchers don’t blindly use w=7 (this is probably not strong enough: “Therefore, these strategies can be very sensitive to the tuning of this parameter and to the choice of the subspace.”)
  • I don’t follow the practical next step from the statement: “For instance, for drift correction, the best in situ calibration strategy can be identified for a considered sensor network knowing its metrological properties and its environment of deployment.” This seems very general and not helpful if I was going to have a sensor deployment where I was limited to where I could deploy sensors.
  • After 36 pages, it would help to have a list of the aspects in this sentence: “Based on a case study, we illustrated all the aspects that should be considered when carrying out this evaluation.”, i.e., 1, 2, 3…
  • Following on that, I find the discussion section 7 useful but section 8 conclusions less so. What practical three to five steps can I learn from this work should I be deploying sensors in an urban setting?

Reviewer 3 Report

The authors should carefully re-read all the manuscript, because some sentences need to be revised and there are some typos.

The manuscript is very long and the authors might consider to shorten some sections. Probably because of the length of the manuscript, the authors have fragmented the article in many sections, subsections, subsubsections,... my feeling is that sometimes the fragmentation is excessive. For example I suggest removing the conclusion subsection in Section 2, as this section is short and does not need a conclusion paragraph.

I am not an expert in the field of environmental sensors, but it is not clear to me which is the innovative part of this paper. My understanding is that the authors propose a reasonable, but super general, framework to test in situ calibration algorithms for environmental sensor networks and they illustrate some case studies. The authors should clarify how the framework they propose differs from the methods used by other literature studies that test calibration algorithms in simulation. 

In section 3.1 the authors list some advantages of simulation-based strategies. They should also comment that simulation approaches require validated models of the measured quantity and the measuring sensor.  

When testing calibration algorithms in simulation, the crucial point is to rely on realistic and validated models of both the measured quantity and the sensor. Indeed the authors showed that the use of different models can remarkably change the results when comparing different algorithms. What are the authors' recommendations about model choice? Are the drift models proposed in section 6.3.1 validated? 

There is a double section "Configuration od the strategies" at page 13.

Round 2

Reviewer 3 Report

The authors have addressed all my comments. The quality of presentation has been improved.